# Early excitatory-inhibitory cortical modifications following skill learning are associated with motor memory consolidation and plasticity overnight

Tamir Eisenstein [1] ✉, Edna Furman-Haran[2] & Assaf Tal [1] ✉

Consolidation of motor memories is vital to offline enhancement of new motor skills and involves short and longer-term offline processes following learning. While emerging evidence link glutamate and GABA dynamics in the primary motor cortex (M1) to online motor skill practice, its relationship with offline consolidation processes in humans is unclear. Using two-day repeated measures of behavioral and multimodal neuroimaging data before and following motor sequence learning, we show that short-term glutamatergic and GABAergic responses in M1 within minutes after learning were associated with longer-term learning-induced functional, structural, and behavioral modifications overnight. Furthermore, Glutamatergic and GABAergic modifications were differentially associated with different facets of motor memory consolidation. Our results point to unique and distinct roles of Glutamate and GABA in motor memory consolidation processes in the human brain across timescales and mechanistic levels, tying short-term changes on the neurochemical level to overnight changes in macroscale structure, function, and behavior.

Learning of a new motor skill does not end when the practice is over. Following practice, offline neural consolidation processes take place to establish the fidelity of the newly encoded motor memory trace[1]. Previous studies have suggested that the consolidation of new motor memories involves a mixture of offline processes taking place at different timescales during wakefulness and sleep[1,2]. For example, while the stabilization of a newly learned skill is usually achieved over the first few hours following practice[2–4], offline enhancement of skill performance (i.e., offline learning gains) following explicit/intentional skill learning usually develops across sleep[1,2]. Both short and longer-term learning-induced functional and structural changes have been demonstrated following motor learning[5–7].

One brain region that has been critically implicated in the consolidation of new motor skill memories is the primary motor cortex (M1)[8], and motor skill consolidation has been shown to involve the formation of a stable neural representation of the acquired skill in M1, termed a motor memory "engram"[9]. Motor learning and consolidation have also been associated with learning-induced plasticity in M1 on multiple levels[9–11], ranging from microscale remodeling of dendritic spines to macroscopic changes in gray matter (GM) volume in humans. Furthermore, changes in the functional communication between M1 and motor learning-related brain regions have also been demonstrated following the acquisition of new skills[5,6]. In addition, offline motor memory reactivation, a putative mechanism of memory consolidation[12], has been demonstrated in M1 in animals following learning[13]. Nonetheless, while the motor cortex has been suggested to play an important role in the early phases of motor memory consolidation[4,14], the mechanisms that support this role, and how they

[1]Department of Chemical and Biological Physics, Weizmann Institute of Science, Rehovot, Israel. [2]Life Sciences Core Facilities, Weizmann Institute of Science, Rehovot, Israel. ✉e-mail: eisentamir@gmail.com; assaf.tal@weizmann.ac.il

relate to longer term learning-induced changes in this region are not clear.

Neural excitation and inhibition (E–I) have been proposed to play an important role in the physiological regulation of cognition and behavior, at both the single neuron level and large-scale circuits[15]. E–I are mediated by the brain's main excitatory and inhibitory neurotransmitters glutamate (Glu) and γ-aminobutyric acid (GABA), respectively. While the physiological balance between excitation and inhibition in the brain is homeostatically regulated, dynamic responses of Glu and GABA induced by external inputs and internal processes have been suggested to be vital to plasticity processes and adaptive behavior following learning[16]. For example, previous works in animals have shown that transient E–I alternations such as GABAergic disinhibition support M1 plasticity[17]. Furthermore, modulation of glutamatergic and GABAergic processing, both in vitro and following learning, has been implicated in the induction of long-term potentiation (LTP), a key mechanism of learning and memory, and the promotion of synaptic strengthening[17,18]. Magnetic resonance spectroscopy (MRS) is currently the only method capable of non-invasively assess excitation and inhibition in the human brain, by

quantifying the concentrations of Glu and GABA. Using MRS, recent studies have provided insights to the role of E–I dynamics in motor learning in humans[6,19]. Yet, at present, most of these studies aiming to understand the neurochemical underpinnings of learning and memory in the human brain have focused on the online phase of learning where the new information is being practiced and encoded. Therefore, we current lack significant understanding of the offline neurochemical responses taking place following learning—especially during the seemingly critical period of early consolidation[2,4]—and how they relate to longer-term learning-induced neurobehavioral changes.

Here, we aimed to take advantage of the increased spatial, temporal, and spectral resolution of ultra-high field 7T MRI and MRS to investigate the hypothesis that early modifications in GABA and/or Glu concentrations following skill learning play an important role in the consolidation of a new motor skill memory as reflected by longer-term overnight improvement in skill performance[1,2] (Fig. 1a, b). Furthermore, by additionally acquiring multimodal structural and BOLD fMRI data, we examined how changes in Glu and GABA concentrations are associated with neural processes thought to support motor skill consolidation such as local and remote functional processing of M1

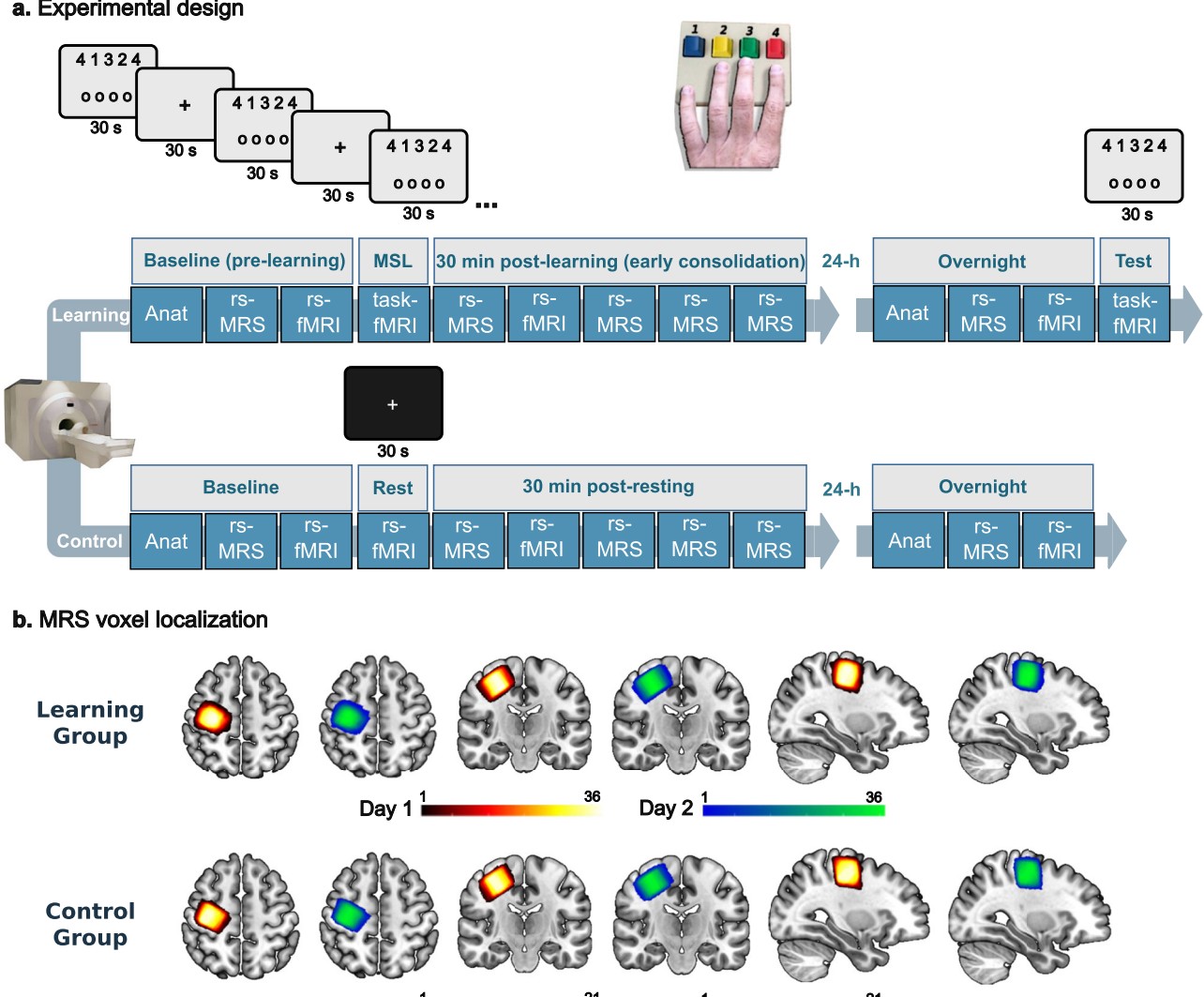

**Fig. 1 | Experimental procedure. a** Schematic illustration of the experimental sessions including neuroimaging experiments and motor learning task/resting control condition. **b** MRS voxel localization during the two scanning sessions across participants and groups—MNI-transformed voxels are presented. Note the consistency in voxel placement across participants, the two experimental sessions, and between the groups. Anat anatomy, MSL motor sequence learning, rs resting-state.

following learning. The consolidation of new motor skill memories has been shown to depend on cortico-striatal brain circuits[8]. The motor cortex serves as a major source of input to the sensorimotor striatum[20]. Cortico-striatal coupling and plasticity over training sessions are essential for the refinement of skilled behavior[8,9,21], and increased coupling between the motor cortex and the striatum following motor skill learning has been shown to occur offline[22]. Moreover, motor cortical-striatal interactions have been suggested to be especially important for the consolidation of the egocentric/movement representation component during motor sequence learning[23]. Previous studies have suggested that striatal structures are differentially engaged at different levels of the learning process, namely that the associative striatum is more involved during the initial stages of skill learning and acquisition, while the sensorimotor striatum which receives inputs from the motor cortex[24] and generally corresponds in primates to the putamen[25], is linked with consolidation and gradual acquisition of behavior[25,26]. Therefore, we examined the relationship between the modifications of Glu and GABA concentrations in M1 shortly after learning, and the inter-regional communication of M1 with the putamen as quantified with resting state fMRI. In addition, we used a multivoxel local correlation pattern (MVLC) analysis, previously used to evaluate memory reactivation in the human brain[27], to determine whether transient changes in Glu and GABA following learning may relate to local M1 functioning following learning as reflected by this correlate of motor memory reactivation. Lastly, since learning-induced plasticity in M1 has been demonstrated to accompany new skill learning, we investigated whether early changes in Glu and GABA concentrations following learning may be related to longer-term structural and functional plasticity in this region, as expressed by overnight changes in M1 GM volume and the functional connectivity with the putamen.

## Results

### No group-level changes in glutamate or GABA following learning

We first measured the temporal dynamics of Glu and GABA levels during the first 30 min following learning or the resting control condition by performing linear mixed-model repeated measures analysis (Fig. 2a). We found a significant main effect of Time for Glu ($F(5,52.6) = 3.96$, $p = 0.004$), with no main effect for Group ($F(1,54.9) = 0.88$, $p = 0.353$). Post-hoc analysis revealed a significant increase during the first post measurement compared to baseline levels (pFDR = 0.005), but not in any other time point compared to baseline (pFDR > 0.288). We did however find a significant Time × Group interaction in Glu dynamics between the groups ($F(5,52.6) = 4.17$, $p = 0.003$), with an unexpected significant Glu increase during the first post measurement in the Control group (pFDR = 0.014), while the Learning group demonstrated a post-learning mean increase which was not statistically significant (pFDR = 0.454). However, between-group post-hoc comparisons showed that the groups were not statistically different in their Glu levels at baseline (pFDR = 0.737), at the first post measurement (pFDR = 0.737) or at any other time point (pFDR > 0.439). It should also be noted that an increase in Glu after 30 min was observed in the Learning group but did not survive the correction for multiple comparisons ($p_{uncorrected} = 0.028$, pFDR > 0.05). We did not find a main effect of Time ($F(5,53.9) = 0.25$, $p = 0.936$), Group ($F(1,54.7) = 0.09$, $p = 0.768$), or Time × Group interaction ($F(5,53.9) = 0.83$, $p = 0.531$) in GABA levels dynamics. In addition, changes were also not observed following learning in other major or Glu-like minor metabolites, i.e., NAA ($p = 0.119$) and Gln ($p = 0.602$), respectively (see Supplementary Fig. 2).

Next, we measured the temporal dynamics of the correlation between Glu and GABA levels following learning compared to pre-learning levels, as a reflection of the balance between excitation and inhibition across participants prior-to and following learning/resting control condition. While a weak, non-significant correlation between

Glu and GABA levels at baseline was demonstrated in the Learning group ($r = 0.182$, pFDR = 0.289), the relationship between the two metabolites significantly increased ($Z = 2.26$, $p = 0.024$) during the post-learning period ($r = 0.582$, pFDR = 0.011) (Fig. 2b). There was no significant change in the correlation between Glu and GABA between baseline and the average post measurements in the Control group ($Z = 1.34$, $p = 0.181$).

We then examined whether the extent of metabolite concentration change after learning/resting condition was related to pre-learning baseline levels. We found a significant inverse relationship between pre and average post Glu levels in the learning group ($r = -0.465$, pFDR = 0.008), but not in the Control group ($r = -0.185$, pFDR = 0.449). However, these correlation coefficients were not statistically different ($Z = 1.03$, $p = 0.304$). Furthermore, we found significant negative correlations between baseline GABA levels and the change in GABA during the post period in both the Learning ($r = -0.677$, pFDR = 0.004) and Control ($r = -0.661$, pFDR = 0.006) groups, which was not statistically different between the groups ($Z = -0.03$, $p = 0.973$) (Fig. 2c).

Lastly, we found no evidence in the Learning group for a relationship between the immediate or averaged changes in either Glu or GABA following learning and the extent of neuro-behavioral activation as reflected by the overall number of key presses performed during the MSL task (immediate Glu change: $r = 0.002$, pFDR = 0.990; immediate GABA change: $r = 0.091$, pFDR = 0.805; averaged Glu change: $r = -0.015$, pFDR = 0.990; averaged GABA change: $r = 0.136$, pFDR = 0.805) or to the M1 engram cluster's activation level during the MSL task (as reflected by the β-value of the practice blocks' regressor in the fMRI GLM analysis) (immediate Glu change: $r = -0.119$, pFDR = 0.805; immediate GABA change: $r = 0.180$, pFDR = 0.805; averaged Glu change: $r = -0.114$, pFDR = 0.805; averaged GABA change: $r = 0.115$, pFDR = 0.805).

### Increased Glu is associated with behavioral improvement overnight

Significant offline learning gains were observed at the group-level, expressed as overnight improvements in skill performance between the last practice block on day 1 and the testing block on day 2 ($F(1,35) = 41.12$, $p < 0.001$), which corresponded to $10.46 \pm 1.8\%$ (mean ± SE) increase in performance (Fig. 3a). Moreover, while most participants demonstrated offline learning gains overnight ($n = 27$), some participants did demonstrate declined ($n = 3$) or levelled ($n = 6$) performance between the two sessions. Next, we examined whether post-learning changes in Glu or GABA could have explained the extent of overnight behavioral learning gains and changes in skill performance. We found that post-learning increases in Glu were predictive of overnight performance improvements when measured immediately following learning ($r = 0.361$, pFDR = 0.048), after 30 min ($r = 0.352$, pFDR = 0.048) and when averaged across the 30 min ($r = 0.472$, pFDR = 0.018) (Fig. 3b). In contrast, changes in GABA levels after learning were not related to overnight offline learning gains (immediate: $r = 0.171$, pFDR = 0.513; after 30 min: $r = 0.095$, pFDR = 0.605; 30-min average: $r = 0.247$, pFDR = 0.513) (Fig. 3c).

### Glu or GABA changes are not related to post-learning functional connectivity changes

Next, we examined whether post-learning changes in either Glu or GABA could explain short-term changes in the functional connectivity between M1 and the putamen, a key region in motor skill consolidation[8]. In this analysis we have focused only on the immediate changes in Glu and GABA in M1 following learning, which preceded the resting-state scan, in order to follow the rationale of a temporal relationship. Immediate changes in either Glu or GABA following learning did not correlate with changes in M1 connectivity with either the right putamen (Glu: $r = 0.002$, pFDR = 0.992; GABA: $r = -0.197$, pFDR =

**a.** Neurochemical dynamics following learning/rest

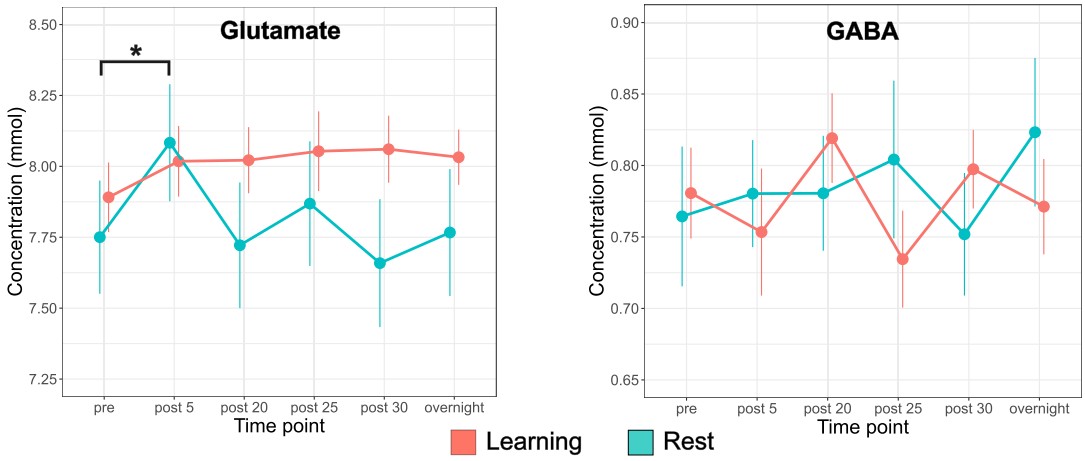

**b.** Increased Glu-GABA coupling following learning

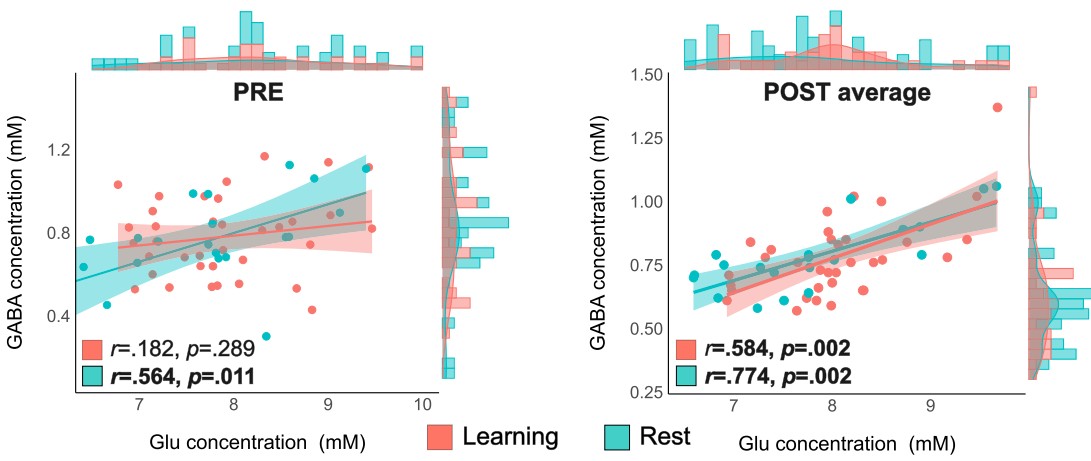

**c.** Post-learning neurochemical dynamics depend on baseline metabolites levels

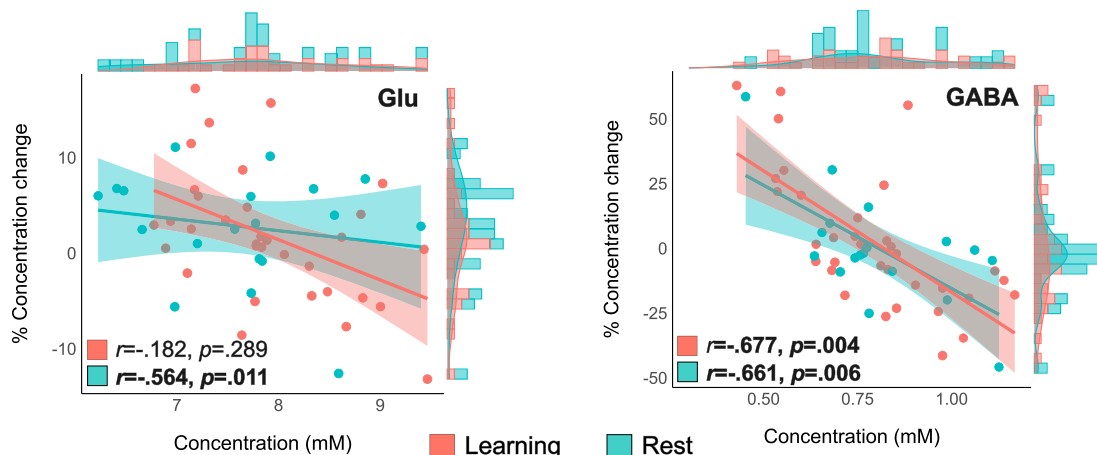

**Fig. 2 | Neurochemical temporal dynamics following motor skill learning and resting control condition. a** Glutamate and GABA dynamics following learning as reflected by changes in concentrations during consolidation and overnight and evaluated with repeated-measures linear mixed models (*n* = 57 participants, means ± SEM are presented, * indicates significant Time × Group interaction *F*(5,52.6) = 4.17, *p* = 0.003). **b** Temporal dynamics of the "balance" between excitation and inhibition in M1 across participants (assessed with Pearson's correlation,

two-sided test) following learning or rest (shaded area around fit line represents 95% confidence intervals). **c** Relationship between baseline metabolite concentrations and the extent of concentration change following learning or rest as measured with two-sided Pearson's correlation (shaded area around fit line represents 95% confidence intervals). *P*-values are corrected for multiple comparisons using FDR. Pink represents the Learning Group and Cyan the Control Group in the plots. Source data are provided as a Source Data file.

**a.** Changes in skill performance during initial learning session and overnight

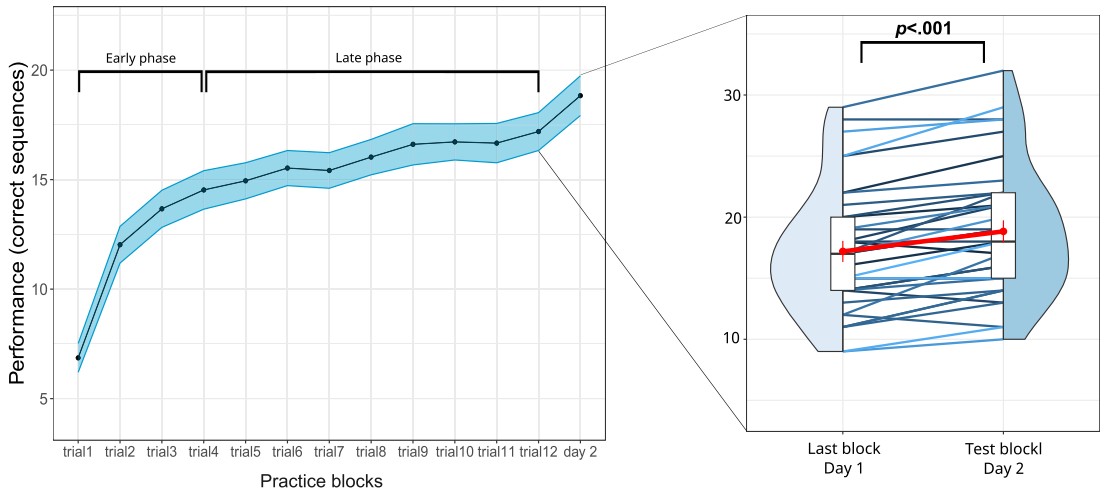

**b.** Post-learning **Glu** changes and overnight changes in skill performance

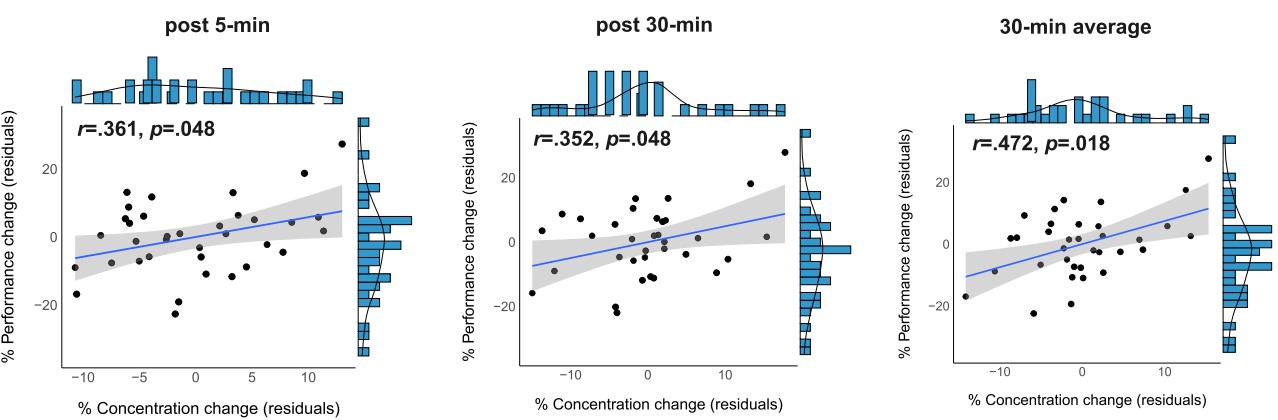

**c.** Post-learning **GABA** changes and overnight changes in skill performance

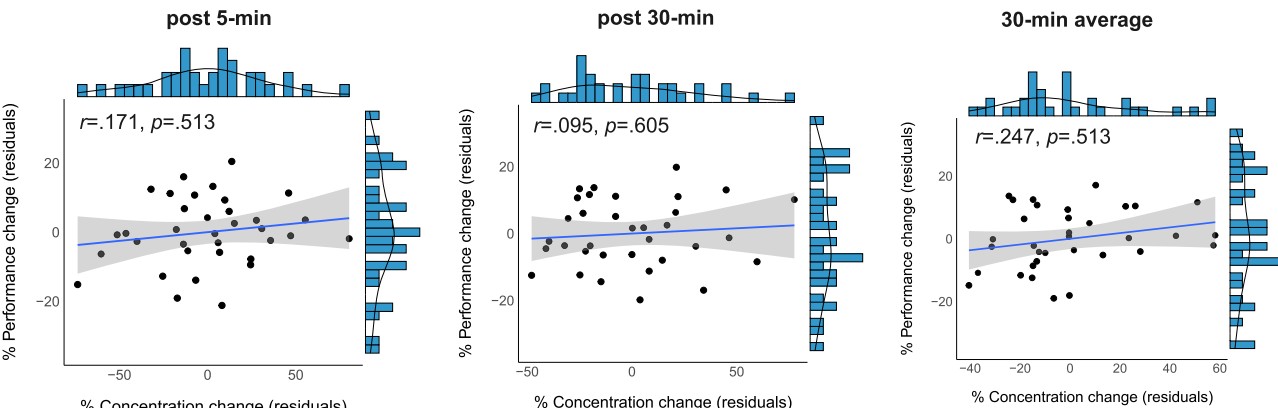

**Fig. 3 | Neurochemical dynamics following learning and behavioral performance. a** Behavioral changes in skill performance during the learning task on the first day (left panel) and overnight (left panel, red dots and vertical lines in the right panel represent mean performance ± SEM on that day, and the blue color coding is used to discern between participants' individual data). Repeated measures linear mixed model analysis revealed significant overnight improvement in performance at the group-level ($n = 36$ participants of the Learning Group, $F_{1,35} = 41.12$, $p = 2.219e$ −07); Immediate, post 30 min, and average changes of Glu (**b**), but not GABA (**c**), following learning are predictive of offline learning gains overnight (two-sided partial Pearson's correlations, FDR-corrected). Shaded area around fit line in panels **b**, **c** represents 95% confidence intervals. Source data are provided as a Source Data file.

0.530) or left putamen ($r = -0.031$, pFDR = 0.992; GABA: $r = -0.291$, pFDR = 0.380). Also, a group-level analysis did not reveal significant differences in functional connectivity of M1 following learning with either the right (right putamen: $F_{(2,35)} = 1.51$, $p = 0.141$) or left ($F_{(2,35)} = 1.39$, $p = 0.247$) putamen.

## Increased Glu is associated with overnight M1 functional plasticity

While we did not observe a relationship between post-learning changes in Glu or GABA in M1 and changes in the functional connectivity between M1 and the putamen shorty following the learning task, when we examined the relationship between post-learning neurochemical changes and overnight changes in the M1-putamen communication, we found increased connectivity between M1 and the right putamen to associate with immediate ($r = 0.368$, pFDR = 0.048) and averaged ($r = 0.373$, pFDR = 0.048) increases in Glu following learning (Fig. 4b). Such a relationship was not evident when Glu changes 30 min following learning were examined ($r = 0.196$, pFDR = 0.275). We did not find a relationship between immediate ($r = -0.270$, pFDR = 0.752) or averaged ($r = -0.085$, $p = 0.752$) Glu changes and overnight connectivity changes between M1 and the right putamen among the Control group. Between-group correlations comparison revealed that the correlation between immediate Glu changes and overnight connectivity changes with the right putamen was significantly higher in the Learning group compared to the Control group ($Z = 2.11$, $p = 0.035$). Furthermore, there was no relationship between immediate ($r = -0.255$, pFDR = 0.438) or averaged ($r = -0.084$, pFDR = 0.642) Glu changes and overnight changes in M1-PCC connectivity. Moreover, within-group correlations comparisons in the Learning group revealed that the correlation between immediate ($Z = 3.42$, $p < 0.001$) and averaged ($Z = 2.31$, $p = 0.021$) Glu changes and overnight connectivity changes with the right putamen were significantly higher compared to the correlations with overnight changes in M1-PCC connectivity.

We did not find significant correlations between changes in connectivity with the right putamen and GABA changes following learning, when measured immediately ($r = -0.141$, pFDR = 0.426), 30 min following learning ($r = 0.248$, pFDR = 0.246), or when averaged GABA changes were examined ($r = 0.300$, pFDR = 0.246). We also did not find a relationship between Glu or GABA changes immediately following learning (Glu: $r = 0.130$, pFDR = 0.473; GABA: $r = -0.074$, pFDR = 0.677), after 30 min (Glu: $r = 0.129$, pFDR = 0.473; GABA: $r = 0.310$, pFDR = 0.176), or averaged (Glu: $r = 0.197$, pFDR = 0.473; GABA: $r = 0.278$, pFDR = 0.176) and overnight changes in the connectivity with the left putamen.

While we did not observe significant overnight connectivity changes at the group-level following learning (right putamen: $F_{(2,35)} = 0.73$, $p = 0.470$; left putamen: $F_{(2,35)} = 0.35$, $p = 0.727$), overnight increases in the connectivity of M1 with the right putamen at the subject-level were associated with overnight offline learning gains and greater improvements in skill performance ($r = 0.308$, $p_{uncorrected} = 0.031$), but did not survive correction for multiple tests (pFDR = 0.144).

## Decreased GABA is associated with overnight increase in M1 GM volume

We then examined the relationship between post-learning neurochemical modifications and overnight changes in M1 GM volume as a marker for learning-induced structural plasticity. A statistically significant correlation was found between GABAergic decreases after 30 min and M1 volume increase overnight ($r = -0.481$, pFDR = 0.015) (Fig. 5), but not with immediate ($r = 0.012$, pFDR = 0.949) or averaged changes in GABA ($r = -0.346$, pFDR = 0.080). Post-learning changes in Glu, either immediately ($r = -0.182$, pFDR = 0.312), following 30 min ($r = -0.266$, pFDR = 0.312) or averaged ($r = -0.226$, pFDR = 0.312) were not significantly associated with overnight changes in M1 GM volume. The correlation between GABAergic changes after 30 min and M1

volume changes among the Control group was not significant ($r = 0.163$, pFDR = 0.562) but significantly different from the correlation observed in the Learning group ($Z = 2.01$, $p = 0.049$). Furthermore, we did not find a relationship between GABA changes after 30 min and overnight volume changes in the PCC among the Learning group ($r = -0.106$, pFDR = 0.557), a correlation that was also significantly different from the one observed between GABA changes and M1 GM volume changes overnight ($Z = 3.15$, $p = 0.002$).

Since the tissue-correction of neurochemical changes and the structural measures were based on the same T1-weighted images, we also examined the correlation between overnight structural changes and the GABA changes following 30 min (the only significant result observed) without tissue correction and also using a Cr-referenced measure. Both analyses yielded significant correlations, similar to the analysis performed with the tissue-corrected values (see Supplementary Fig. 3).

While we did find a link between overnight changes in M1 GM volume and post-learning neurochemical modifications at the subject-level, there was no group-level change in M1 GM volume across the two days following learning ($F_{(1,35)} = 1.59$, $p = 0.215$)), and the extent of overnight M1 structural changes was not related to overnight changes in task performance ($r = -0.115$, pFDR = 0.345).

## Increased Glu is associated with greater MVLC similarity after learning

Lastly, after establishing a connection between neurochemical and overnight neurobehavioral changes, we examined whether changes in Glu and GABA following learning were associated with a putative offline consolidation-related function, i.e., motor memory reactivation in M1 (Fig. 6a). Group-level repeated measures mixed model analysis did not reveal a significant difference between the similarity of the post-learning resting MVLC patterns with the task MVLC patterns compared to the pre-learning rest patterns ($F_{(1,35)} = 0.03$, $p = 0.876$). Thus, at the group level we did not find evidence for offline motor memory reactivation based on the MVLC patterns. We next examined whether MVLC pattern similarity difference could be related to the post-learning changes in Glu and GABA on a subject-by-subject basis. In this investigation we have focused on the first (immediate) post-learning MRS measurement which directly preceded the post-learning resting-state scan in order to follow a theoretical basis for temporal relationship between the two phenomena. We found that greater increases in Glu immediately following learning were associated with higher similarity of post-learning resting patterns with the task compared to pre-learning patterns was ($r = 0.369$, $p = 0.035$) (Fig. 6b). Greater reductions in GABA immediately after learning were not significantly related to higher similarity of the post-learning resting patterns with the task ($r = -0.315$ $p = 0.074$). While memory reactivation has been proposed as a strong mechanism of memory consolidation[12,28], we did not observe a significant relationship between MVLC similarity difference and overnight changes in skill performance ($r = 0.201$, pFDR = 0.256).

## Discussion

### Lack of group-level changes in Glu or GABA following learning
The aim of the current study was to explore the dynamics of Glu and GABA in M1 during the early stages of motor memory consolidation, and how these relate to neuro-behavioral plasticity following MSL. We examined how early changes in Glu and GABA following learning relate to overnight changes in skill performance, inter-regional functional processing of M1, and to structural changes in this region (see Supplementary Fig. 4 for a summary of the significant associations observed between Glu or GABA changes and neuro-behavioral correlates of motor memory consolidation). These questions follow previous findings suggesting that the initial hours after skill learning are vital for the consolidation of motor skills[2]. Interestingly, however, at

**a.** ROIs localization

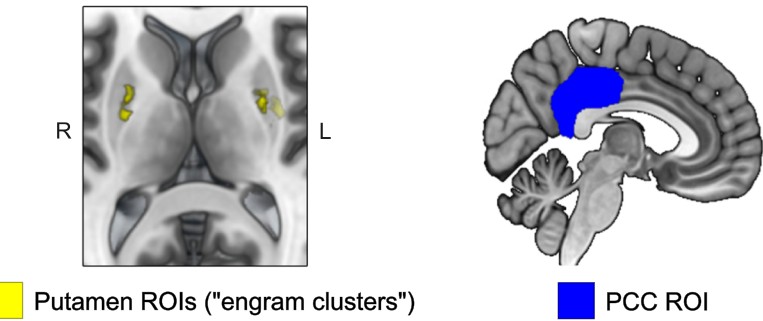

☐ (yellow) Putamen ROIs ("engram clusters")   ☐ (blue) PCC ROI

**b.** Increased Glu following learning, but not rest, is associated with increased FC of M1 with the R putamen, but not with the PCC

**M1 - R putamen (5 min post)**

*r*=.368, *p*=.048 (Learning)
*r*=-.270, *p*=.752 (Rest)

FC difference (residuals) vs % Concentration change (residuals)

**M1 - R putamen (30 min average post)**

*r*=.373, *p*=.048 (Learning)
*r*=-.085, *p*=.752 (Rest)

☐ Learning   ☐ Rest

**M1 - PCC (5 min post)**

*r*=-.255, *p*=.438

**M1 - PCC (30 min average post)**

*r*=-.084, *p*=.642

**Fig. 4 | Glu changes and M1 functional connectivity with the putamen.**
**a** Bilateral putamen and PCC ROIs definition. **b** The relationship between immediate and averaged changes in Glu following learning/rest and overnight changes in the functional communication between M1 and right putamen, and between M1 and the PCC following learning measures with two-sided partial Pearson's correlation, FDR-corrected (shaded area around fit line represents 95% confidence intervals). Pink represents the Learning Group and Cyan the Control Group. FC functional connectivity, L left, R right. Source data are provided as a Source Data file.

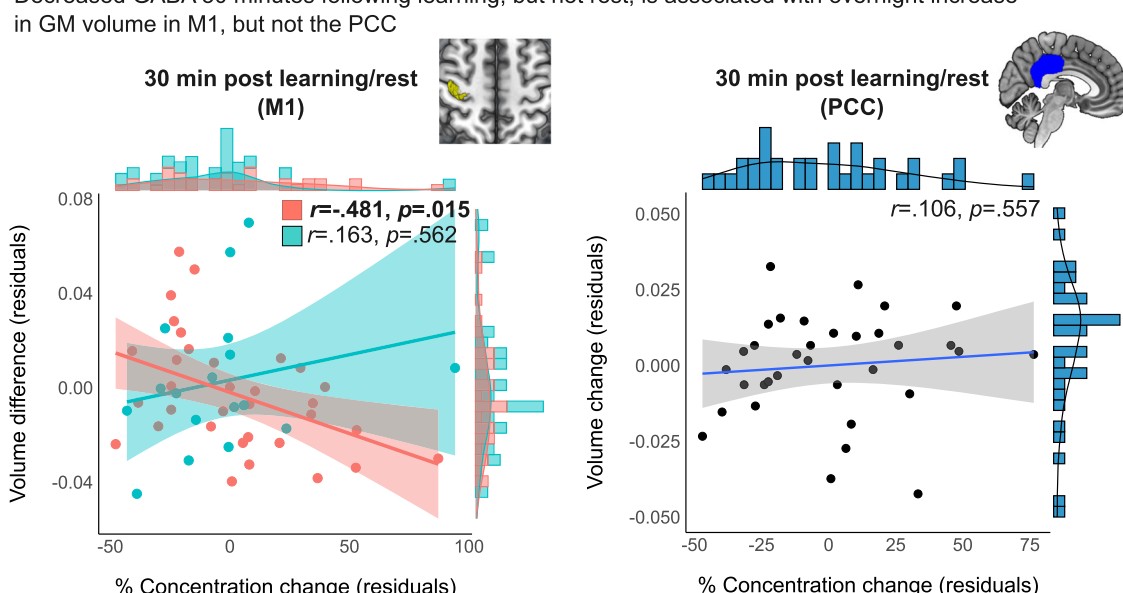

**Fig. 5 | GABAergic changes following learning/rest and overnight changes in M1 or PCC GM volume.** Structural changes and GABAergic reductions 30 min following learning is predictive of increased M1 GM volume overnight, but not following rest or in the PCC. *Z* scores and *p*-values represent the result of two-sided comparisons of the partial Pearson's correlation coefficients using r-to-z transformation between groups (left panel) or within learning group (right panel). Shaded area around fit line represents 95% confidence intervals. Pink represents the Learning Group and Cyan the Control Group. Source data are provided as a Source Data file.

the group-level, we did not find significant changes in Glu or GABA, either immediately or during the first 30 min following the task. This is partially in contrast to a recent study by Maruyama and colleagues[5] which found decreased Glu immediately after learning. However, the reduction in Glu observed in their study was similar in both the learning group and a non-learning control group, suggesting that it did not represent a specific learning-induced physiological response. GABAergic inhibitory neurons display wide variety of morphology and physiological properties, and different GABAergic neurons subtypes target different anatomical domains of excitatory neurons, enabling them to regulate different aspects of the spatiotemporal activity of the glutamatergic cells[29]. Therefore, GABAergic modulation following learning may be subtype-specific and function-specific, as was evidence by both increased and decreased inhibitory axonal boutons among different GABAergic cells subtypes following the same motor learning[30]. This opposite modulatory pattern may in turn underlie a lack of absolute mean change in GABAergic concentration across the motor cortex, and may explain why a group-level change in whole M1 GABA was not observed with MRS following motor learning in the current and other studies[5]. In contrast, the subject-based correlation we did find in the current study between GABA dynamics and other neural metrics may reflect the dynamics within the GABAergic pool itself across different individuals. Furthermore, the observation that the same stimulus (i.e., MSL task) resulted in opposite metabolic responses across different participants, follows previous reports on such an opposite response across different individuals (even if a significant group-level effect was observed) following other types of learning (i.e., perceptual)[31,32] as well as in response to non-invasive brain stimulation in M1[33,34]. Those observations, taken together with the findings of the current study, may suggest that the direction of the neurochemical cortical response to external or internal stimuli across different individuals, may represent an individual trait, which may underlie or contribute to the inter-individual differences in learning capacity and potential. However, this possibility needs further direct empirical examination.

While we did not observe group-level changes in Glu and GABA following learning, we did observe a significant increase in the coupling (i.e., correlation) between the metabolites' concentrations during the post-learning period compared to pre-learning but not following the resting control condition. This increased "coherence" between Glu and GABA levels following learning may represent a more fine-tuned regulation of the balance between inhibition and excitation that may be necessary for further consolidation processes. In addition, it could also reflect greater homeostatic control mechanism aiming to restore the balance between excitation and inhibition after it has been perturbed, for example by learning-induced disinhibition during the online learning experience[19,35]. Nevertheless, both of these hypotheses require further examination. Previous works demonstrated changes in E-I during the online phase of motor learning, mainly expressed as decreased GABAergic levels[19,35]. Although we did not measure changes in Glu and GABA during the online learning phase, it is possible that the increase in Glu–GABA coupling following the task may represent a negative feedback mechanism aiming to restore baseline physiological functions after being perturbed during the online learning phase. On the other hand, as the E–I balance has been proposed to modulate important aspects of cognition and behavior[15], the increased Glu–GABA coupling may also reflect a direct learning-related phenomenon, presumably representing the controlled processing of the new neural representation of the learned motor skill. Furthermore, it was recently suggested that the E–I balance may be vital for neuronal computation that is robust to noise and for shaping efficient neural coding[36,37] and memory states[36]. As motor memories are significantly vulnerable to disruption and interference during the initial offline period immediately following learning, increasing the Glu–GABA coupling may be critical for the stabilization of the new motor memory trace[2,4].

We also found that the extent and direction of GABAergic changes over time were strongly related to baseline GABA levels. Specifically, higher GABA levels at baseline were associated with greater reduction in GABA following either learning or the resting condition, and vice

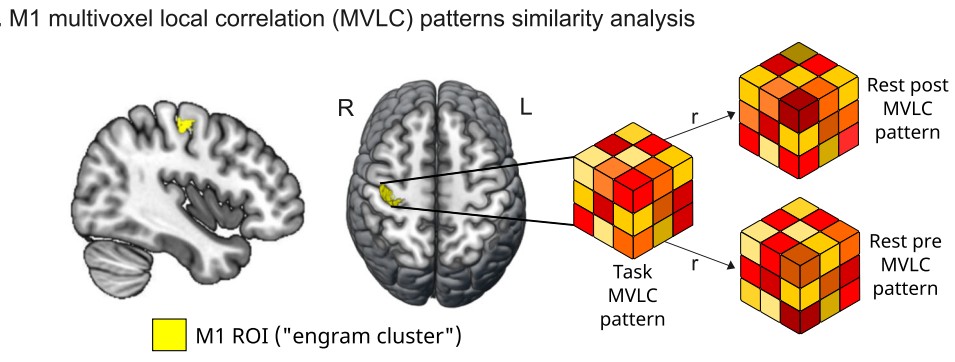

**a.** M1 multivoxel local correlation (MVLC) patterns similarity analysis

M1 ROI ("engram cluster")

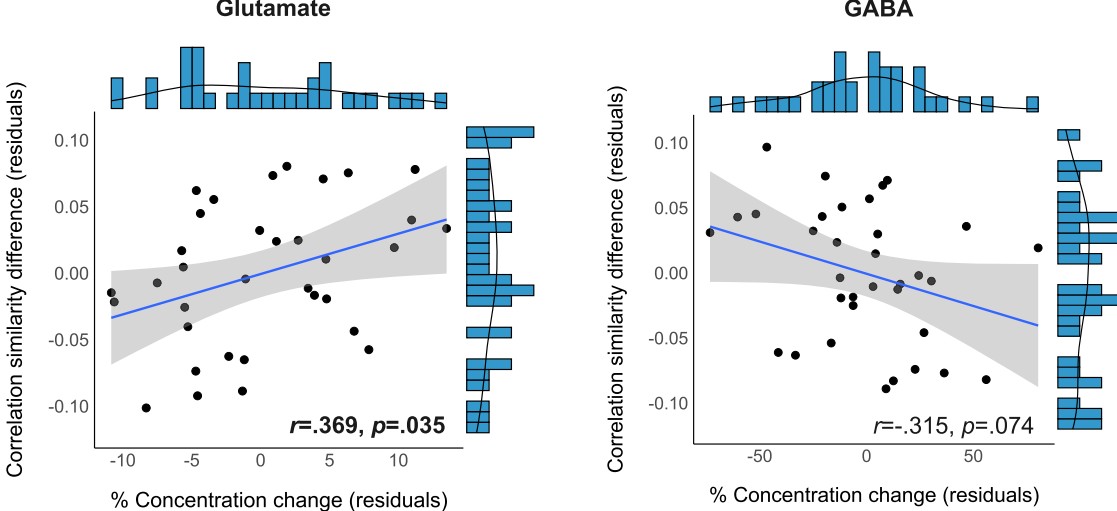

**b.** Neurochemical dynamics following learning are associated with greater MVLC pattern similarity

**Glutamate**

**GABA**

**Fig. 6 | Multivoxel local correlation patterns and neurochemical changes following learning. a** MVLC analysis and M1 ROI definition (yellow) based on activation patterns on day 1 and 2. **b** The relationship between MVLC similarity differences and immediate Glu and GABA changes following learning measured using two-sided partial Pearson's correlation (Shaded area around fit line represents 95% confidence intervals); L left, R right, MV(L)C multivoxel (local) correlation, ROI region-of-interest. Source data are provided as a Source Data file.

versa. This inverse relationship is in accordance with previous findings[38], and is therefore suggestive of a general homeostatic mechanism that act to regulate GABA levels over time, as was also proposed to occur during sleep following learning[39]. In contrast, we found a similar inverse relationship for Glu levels following learning, but not following rest. This in turn, may suggest that short-term Glu dynamics, in contrast to GABA, may be more sensitive and reflective of recent neural responses to external stimuli. However, while it is possible that this inverse relationship also reflects a homeostatic mechanism following activation (i.e., the learning task), we did not find an association between the extent of finger movements or M1 activation during the MSL task and the extent of post-learning changes in either Glu or GABA.

**Early offline Glu changes are associated with motor skill consolidation**

While no group-level change in neurochemical responses following learning was observed in the current study, we did find an association between increased Glu following learning and the extent of overnight improvements in skill performance on a subject-by-subject basis. It is well established that glutamatergic neurons are critically implicated in the induction of LTP, one of the most-studied forms of synaptic plasticity[40]. LTP-induced synaptic modifications have been shown in the rodent M1 following artificial stimulation and motor learning[17,41], and LTP-like plasticity can also be induced in the human M1, as

demonstrated using non-invasive brain stimulation (NIBS) methods such a transcranial magnetic stimulation (TMS) and transcranial direct cranial stimulation (tDCS). This in turn, results in increased M1 motor-evoked potentials which may last from 30 min (early post-stimulation) to hours (late post-stimulation)[42,43]. Moreover, NIBS methods have also been used to provide evidence for the contribution of LTP-like mechanisms to motor learning in the human M1[42], as also measured immediately after initial practice[44]. Therefore, the temporal dynamics of neurochemical changes examined and observed in the current study may presumably reflect an expression of LTP-like processes, as the early phases of LTP are expressed within the first minutes to hours following learning or appropriate stimulation, and are dependent on post-translational synaptic modifications[45].

While LTP-dependent synaptic strengthening has traditionally been associated with post-synaptic modifications of Glu receptors (namely, AMPA and NMDA), LTP-dependent increased synaptic efficacy has also been proposed to be expressed pre-synaptically by an increase in the release of Glu by the pre-synaptic neuron, resulting in increased amount of transmitter outside the vesicles[46–49]. Such an effect was documented in the rodent M1 30 min following motor learning, and was associated with increased neuronal excitability[50]. It has been proposed that the vesicular pool may be largely invisible to MRS compared to the extracellular and cytosolic pools of neurons and astrocytes, and that dynamic changes in the MRS Glu signal may arise (at least partially) from shifts between compartments that may take

place during increased neuronal activity[51,52]. Furthermore, this effect was suggested to be more pronounced when MRS is measured with intermediate and longer TE (30 ms or higher, as also utilized in the current experiment), due to faster T2 relaxation rate of restricted Glu within the vesicles[53]. Therefore, a presynaptic LTP-related shift of more Glu into MR-visible compartments (i.e., extra-vesicular) may underlie the positive association between increased Glu observed following learning and overnight skill performance, similar to what have been proposed to occur during task-related functional activation[51]. This hypothesis, however, may need further examination, preferably using the combination of MRS with invasive brain stimulation in animal models. In addition, it should be noted that ¹H-MRS does not enable to distinguish between intra-synaptic, extra-synaptic or intracellular compartments, or between different cell types. Therefore, while Glu subserves physiological functions in both neurons and astrocytes and plays a dual role in excitatory neurotransmission and cellular energy metabolism, the origin and physiological pathways of the MRS Glu signal are currently could not be discernible using ¹H-MRS. Therefore, Glu concentration changes may also reflect increased anaplerotic reactions subserving the tricarboxylic acid cycle (TCA) cycle during or following increased neural activity. Furthermore, there is a well-documented coupling between oxidative metabolism (and the potential energetic role of Glu) and the glutamate-glutamine cycling[54]. While a compartmental shift mechanism based on the visibility of cellular compartments to MRS may support the contribution of increased vesicular release and changes in neurotransmission to the dynamic changes observed in Glu, it needs further investigation. Nevertheless, although we cannot point out to the physiological origin of the increased Glu associated with motor memory consolidation in the current study, either increased neurotransmission or increased metabolic activity could reflect elevated levels of neural processing following learning that are bound to the early stages of offline consolidation processes.

Interestingly, while we did not observe significant changes in M1 neurochemistry at the group-level, we did find significant group-level increases in skill performance overnight. Skill enhancement following explicit MSL paradigms has been proposed to depend on post-learning sleep to develop[1,2], possibly due to post-learning inhibitory mechanisms preventing offline learning gains to develop over wakefulness and presumably delaying it until sleep-dependent consolidation processes are initiated[55]. However, it was previously shown that when this decrease in M1 excitability immediately following explicit skill learning is prevented by increasing cortical excitability, offline learning gains that would not normally occur over wakefulness can be induced[56]. Furthermore, significant correlation was found in that study between the extent of cortical excitability and subsequent offline improvements, highlighting the importance of M1 excitability immediately following explicit learning, and causally linking it to motor memory consolidation. Robertson and colleagues (2005)[14] have demonstrated that applying TMS to disrupt M1 function following implicit motor skill learning blocked offline learning gains from developing over the day but not overnight. Furthermore, this effect was both spatially and temporally specific, as applying the TMS a little anterior to M1, or 2 h after learning instead of immediately following it, did result in significant learning gains over the day. In addition, Breton and Robertson[57] showed that disrupting the function of M1, but not inferior parietal lobule, immediately following explicit MSL prevented offline learning gains from developing over a night of sleep. Taken together, those findings suggest that M1 may be critical to the development of offline learning gains following explicit MSL during both wakefulness and sleep. This in turn, may explain how significant group-level increase in skill performance was observed overnight despite a lack of group-level changes in Glu or GABA following the MSL paradigm. As suggested by Robertson and colleagues[14], skill enhancement may not be a single neural process and instead different neural mechanisms

with differential dependencies on M1 may promote offline improvements over different brain states (i.e., wakefulness and sleep). Hence, the correlation we found between increased Glu shortly following learning and overnight skill performance, may reflect an additive effect of offline learning gains accumulating over the day and overnight. Further research applying differential modulation of M1 function during wakefulness and sleep following explicit MSL is needed in order to establish the feasibility of such an additive effect.

### Increased Glu relates to M1 function and plasticity following learning

Increased cortical excitability has been shown to affect neural processes that are essential to the development of offline behavioral improvements and cortical plasticity. One of these processes may be the offline reactivation (i.e., replay) of recently acquired motor memories. Offline memory reactivation is currently thought to play a vital role in the consolidation of explicit and implicit memories[12]. While offline memory reactivation has been investigated mainly in the hippocampus and in the context of declarative memory, recent evidence suggest that a similar process occurs in the motor cortex in both animals[13] and humans[58] following motor learning. Here, we found that increased Glu (and to a lesser extent GABAergic reduction) were associated with higher similarity of MVLC resting patterns in M1 immediately following learning to the patterns observed during the learning experience itself, potentially reflecting a process of offline motor memory reactivation. Intra-regional patterns of multivoxel functional connectivity were previously used to examine offline memory reactivation in humans[27], although these studies have focused on other brain regions (e.g., the hippocampus). Furthermore, although the current study did not utilize a direct measure of neuronal reactivation/replay, the recent evidence for a motor memory-related replay in M1 at the cellular level following motor learning in rodents[13] may support the feasibility of this process in the human motor cortex as well[58].

In addition to the well-established role of glutamatergic modifications in functional synaptic plasticity, GABAergic modulation has also been implicated in these processes following learning. Evidence from the animal literature have demonstrated GABAergic disinhibition to be essential for the induction of LTP-like plasticity and the facilitation of LTP-like synaptic responses in M1[18]. Animal studies have also shown motor cortical disinhibition at the cellular level during and following motor practice that is expressed by rapid elimination of inhibitory boutons on layer II/III excitatory glutamatergic neurons[30] and decreased release of GABA[17]. Previous studies in humans have demonstrated a potential link between local GABA levels and functional connectivity of the motor cortex. Specifically, lower GABA concentrations were found to associate with increased resting state functional connectivity of the motor network[59], while decreased GABA-to-Glu ratio (i.e., disinhibition) was associated with greater increases in M1 functional connectivity with the frontoparietal network following motor learning[5]. Here, however, we found a relationship between short-term Glu increases following learning and overnight increases in M1 connectivity with the putamen, with no significant association with GABAergic modifications following learning. The consolidation of motor skill memories has been demonstrated to depend on motor cortical–striatal interactions which support the development of the egocentric/movement representation component of new motor sequence learning[8,9,21,23,60]. Motor learning was shown to induce strengthening of M1 engram neurons' synaptic outputs to the striatum, which may be modulated by changes in the neuronal output of M1 excitatory cells at striatal dendrites[9]. However, these modifications in turn would have been neurochemically expressed remotely of the motor cortex, and therefore cannot directly explain the relationship observed in the current study between Glu increase at the site of M1 and increased M1-putamen connectivity. With that being said, it is

important to note that such a relationship was observed with respect to an MSL-related region (i.e., the putamen) and not with the PCC, which is not generally implicated in learning new motor skills[8,61]. This discrepancy in turn support specificity in the observed effect of Glu changes to the motor skill learning process. Furthermore, it was also shown that cortico-striatal plasticity keeps developing between training sessions during the motor learning process[21,60], suggesting of a continuous re-organization of the neural representation of the skill on a longer time-scale, and not necessarily during initial practice or immediately following it. While it is now generally accepted that post-learning sleep plays a role in motor memory consolidation following explicit skill learning[23], two previous studies in both rats[22] and humans[62] provided evidence that learning-related modulation of cortico-striatal activity take place during sleep, and that particularly non-rapid eye movement (NREM) sleep and sleep spindles are essential in shaping this connection. Lemke and colleagues (2021)[22] demonstrated that increases in functional connectivity between neurons in M1 and in the dorsolateral striatum (generally homologs to the putamen in humans[25]) occurred offline following learning, and not during learning itself as measured immediately after practice. Boutin and colleagues (2021)[62] further found using electroencephalography (EEG) that greater spindle band coherence between M1 and the putamen in the contralateral hemisphere to the training hand was related to overnight improvements in skill performance. Our results may complement those findings as we found short-term glutamatergic changes in M1 to associate with overnight increases in the functional connectivity between M1 and the putamen, but not shortly after the learning session. Moreover, this relationship was specific to the contralateral hemisphere to the learning hand (i.e., right M1 and putamen), which by itself was related to greater improvements in performance, regardless of neurochemical changes following learning (although did not survive multiple comparisons correction). Furthermore, while Lemke and colleagues[22] found that offline increases in cortico-striatal connectivity correlated with skill performance, they also reported that only 35% of M1 and striatal electrode pairs that were examined demonstrated offline increases in connectivity, with the others electrode pairs exhibiting either decreases or no change. This may support our results as we did not find significant overnight changes in M1-putamen connectivity at the group-level following learning.

## Offline reductions in GABA relate to overnight M1 GM volume changes

In addition to learning-induced functional synaptic strengthening, synapses also store information by modifying their structure. Learning-induced structural changes have been shown to be expressed with functional specificity such as the differential formation and stabilization of new spines in M1 engram cells[9]. Accumulating evidence from animals and humans suggest that structural changes following motor learning may be rapidly induced and are already evident shortly after the learning experience. Xu and colleagues[63] reported rapid increase (within an hour) in spine density within the motor cortex of mice following skill learning, which returned to baseline only several days later. Chen and colleagues[30] demonstrated significant spine increases in the mouse M1 within 2 h following motor learning and overnight. In humans, Taubert and colleagues[10] demonstrated learning-specific increased motor cortical thickness 1 h following a single balance training session which could not be explained by changes in resting cerebral blood flow. Furthermore, reductions in mean diffusivity, an MRI-derived correlate of tissue microstructure, were observed in the motor system as early as 45 min following MSL[7]. Interestingly, Kodama and colleagues[11] found significant overnight increases in M1 GM volume following learning which predicted behavioral improvements several days later. Therefore, it is also possible that short-term structural changes at a given time point following motor learning may not necessarily reflect the current state of

learning-induced adaptive behavior but rather constitute an intermediate phase in the overall learning process. Furthermore, since learning-induced structural plasticity involves processes of both formation and elimination of existing synapses that are expressed on different timescales[63], inter-individual variability in the temporal dynamics of these processes may underlie variable expressions of structural plasticity when measured at a given time-point during the learning process.

Dendritic spines remodeling such as enlargement of existing spines or the formation of new ones[40] have been well documented following the induction of LTP with different temporal expressions. For instance, the artificial induction of LTP has been shown to increase the volume of the dendritic pool in layer V of the rat sensorimotor cortex after 15 days[64], while artificially inducing LTP in hippocampal cultures resulted in the formation of new dendritic spines as early as 45 min following the stimulation[65]. Previous studies have demonstrated that intense local release of Glu or GABA can induce post-synaptic dendritic spines formation[16]. By utilizing two-photon uncaging of caged glutamate compounds, which reliably stimulates single spines, it was shown that releasing Glu into the synaptic cleft and maximally activating NMDA receptors induced robust spine enlargement in the adult mouse neocortex[66]. Interestingly, increased release of GABA was shown to induce dendritic spine enlargement only during early life development, when GABA is still an excitatory transmitter, further highlighting the importance of excitation (or disinhibition) to synaptic plasticity following initial learning. This suggest that presynaptic neurotransmitter release may constitute the main trigger for structural synaptic plasticity and highlight the roles of Glu and GABA in promoting synaptic remodeling[16]. Furthermore, reduced short-interval intra-cortical inhibition (SICI), a TMS-derived measure proposed to reflect motor cortical GABA$_A$ inhibitory function, was demonstrated early following initial motor learning[44]. Since GABAergic disinhibition is necessary for LTP in the motor cortex[67], decreases in GABA release probability may promote Glu-mediated plasticity. For instance, Kida and colleagues[50] found a transient decrease in pre-synaptic GABA release probability after 30 min following motor learning in the rat motor cortex that was associated with increased M1 excitability. Furthermore, Chen and colleagues[30] demonstrated significant elimination of GABAergic boutons on excitatory neurons in the mouse M1 within 2 h following motor learning. Interestingly, while we did not observe a group-level change in M1 GABA levels, a correlation with structural modifications was only evident for GABA levels changes expressed 30 min following learning but not immediately following practice.

While structural cortical changes have been demonstrated following motor learning in the human brain, linking those neuroimaging-derived metrics to specific microscopic modifications is not trivial. Also, while the animal literature proposes a strong link between structural plasticity and adaptive behavior following motor learning, findings in humans are less consistent in establishing such as relationship[68], as was also evident in the current study. This in turn may result from the limited mechanistic specificity that is inherent in structural human neuroimaging methods and makes the direct linkage between human and animal findings less straightforward. Investigating structural brain changes in humans is currently limited to the relatively macroscopic neuroimaging voxels, which are composed of a mixture of neural and non-neural components including neurons, glial cells, vasculature and interstitial space, with over 50% of cortical GM estimated to be composed of neuropil (i.e., axonal, dendritic, and glial processes)[69]. Therefore, changes in GM structure as evident with MRI could result from a combination of a variety of processes such as neurogenesis, gliogenesis, synaptogenesis, dendritic and axonal remodeling, cell swelling, and vascular changes. However, while formation of new cells (i.e., neurogenesis or gliogenesis) may require longer timescales, short-term structural changes ranging from hours

to few days following learning may be more related to dendritic spines formation and rapid modification of astrocytes morphology, cellular swelling, and changes in the ratio between intra- and extra-cellular compartments[63,70].

In summary, many human studies to date have focused on the ratio between Glu and GABA as a reference for the balance between excitation and inhibition. Here, we show that Glu and GABA may be independently associated with different aspects of motor memory consolidation and plasticity on different timescales, and that these metabolites may underpin distinctive functions in learning and memory processes. We have demonstrated that changes in Glu and GABA in M1 early following motor learning may be important for offline consolidation processes and the promotion of structural and functional cortical plasticity. Also, while the post-learning neuro-behavioral changes were mostly associated with immediate or averaged post-learning changes in Glu, these were more linked to GABAergic changes only expressed 30 min after learning, suggesting that Glu- and GABA-dependent plasticity processes may operate on different timescales during the early phase of motor memory consolidation. Hence, the current study provides important insights to our basic understanding of the multidimensional mechanisms of learning and plasticity in the human brain. Furthermore, our findings may also have important clinical implications. Different applications of NIBS methods such as TMS and tDCS have been shown to be able to modulate cortical excitability and LTP-like plasticity, and to promote GABAergic and glutamatergic changes. Therefore, the potential key role of early post-learning neurochemical modifications to motor learning and plasticity that was revealed in the current study may be further examined in clinical trials and clinical settings of rehabilitation following stroke or brain injury using the methods mentioned above.

## Methods

### Participants
57 healthy right-handed young adults (age $27.5 \pm 5.2$ years, 24 females) participated in the current study. Sex of participants was determined based on self-report and was not considered as a variable of interest in the study design. All participants provided written informed consent, approved by the Wolfson Medical Center Helsinki Committee (Holon, Israel), and the Institutional Review Board (IRB) of the Weizmann Institute of Science, Israel. Exclusions criteria included age below 18 years or above 40 years, musicians or video gamers (past or present), any neuro-psychiatric history (including medications), and participants who did not meet the safety guidelines of the 7T scanning policy. Participants received monetary compensation for the participation in the study.

### Experimental protocol
We conducted a $2 \times 2$ (between-group and within-subject) repeated measures experiment using a multimodal MR approach implemented on an ultra-high field 7T MRI scanner (Fig. 1a). Participants were divided into a Learning group ($n = 36$, age $27.2 \pm 3.8$ years, 15 females) and a Control group ($n = 21$, age $27.9 \pm 0.7$ years, 9 females), and arrived at the lab at two consecutive days. During the first day, participants underwent anatomical, single-voxel MRS and resting-state fMRI scans prior to and following a motor sequence learning task (i.e., the learning group) or an equivalent resting period. Task-induced BOLD data was recorded during the learning task. On the second day, participants underwent anatomical, resting state fMRI and MRS only once at baseline, prior to a motor learning evaluation fMRI paradigm in the case of the Learning group (i.e., a testing paradigm). Only resting measurements were collected among the Control group's participants on the second day.

### Motor learning task
All participants in the Learning group performed an explicit motor sequence learning (MSL) task in which they were asked to repetitively tap a five-digit sequence (4-1-3-2-4) with their non-dominant left hand, as fast and accurately as possible[71]. Keypresses were performed on an MR-compatible response box with four computer-like pressing keys (Cedrus Corporation, Lumina LS-LINE model). The response box was placed near the left thigh, was adjusted for each participant's arm length, and fixed to this position to prevent its movement during the scan. Keypress 1 was performed with the little finger, keypress 2 with the ring finger, keypress 3 with the middle finger and keypress 4 with the index finger. The task consisted of 12 trials, lasting 30 s each during the first day (of initial learning). Each two consecutive task blocks were separated by a 30 s fixation block in which participants were asked to fixate on a black cross presented at the middle of a bright screen. The second day (of learning evaluation) consisted of a single block of 30 s. The five-digit sequence (4-1-3-2-4) was projected on the screen during each task block. Each task block also included four white circles presented on the screen, and each finger press was followed by a corresponding circle being filled for the time duration of that specific press. This procedure was implemented to provide the participant with online visual feedback (whether pressing on the desired key) and the experimenter with online information regarding the execution of the correct finger sequence. The visual circles did not provide error feedback, only information on the pressing finger at a given time. The first day also included a short pre-scan familiarity session with the task on a lab computer, in which only the experimenter performed a short version of the task with a control sequence for demonstration, to prevent any learning effect in the participant prior to the actual task in the scanner. Therefore, the "real" sequence was only revealed to the participants when starting the learning paradigm. Importantly, all participants demonstrated at least one correct sequence pressing during the first learning block. Behavioral performance in the task was evaluated by the number of correct sequences performed within each block, a common behavioral measure that combines both the speed and accuracy of the performed skill[71]. As motor skill learning is a gradual process characterized by an initial stage of relatively fast performance increase (i.e., early phase), followed by a slower stage of a more gradual performance change over additional practice (i.e., late phase)[72], we have defined these phases of learning during the first day for further analyses which are described below. Importantly, the duration of each stage is highly dependent on the complexity of the acquired motor skill, and both stages may be evident on the first practice session when learning simple skills such as short key-press sequences[58].

Therefore, the late phase of the learning session was defined as the practice block from which no block-to-block significant changes in performance were observed at the group-level ($\alpha < 0.05$). This evaluation was based on non-corrected pairwise comparisons between consecutive blocks following a random-intercepts and random-slopes linear mixed model analysis with participants' ID as the random effect. According to this, the first three blocks comprised the early phase, and blocks 4–12 comprised the late phase. This division is similar to what was reported in a recent study[58] implementing the same learning sequence (when this transition occurred after ~30% of the blocks). Visual stimuli presentation and data collection were conducted using the Psychophysics Toolbox Version 3 (http://psychtoolbox.org/) implemented in MATLAB.

### MRI scanning procedures
**MR data acquisition.** The scanning sessions were performed on a 7T Terra scanner (Siemens-Healthineers, Erlangen, Germany) using a commercial single-channel transmit/32-channel receive head coil (NOVA Medical Inc., Wilmington, MA, USA), capable of maximum $B_{1+}$ amplitude of 25 µT. Soft pads were used to hold each participant's head in place to minimize head movement during the scanning sessions. An initial localizer and gradient echo field map were acquired for automated $B_0$ shimming (Scan parameters for B0 mapping: TR/TE1/

TE2 = 406/3.06/4.08 ms, $\alpha$ = 25°, 1.9 × 1.9 × 2.0 mm resolution, TA = 1:04 min). A high-resolution structural T1-weighted MP2RAGE (Magnetization Prepared 2 Rapid Acquisition Gradient Echoes) image was acquired for voxel placement and subsequent tissue segmentation (TR/TE/TI1/TI2 = 4460/2.19/1000/3200 ms, $\alpha1$ = 4°/$\alpha2$ = 4°, 1 mm³ isotropic voxels, TA = 6:56 min). For the MRS acquisitions, a 2 × 2 × 2 cm³ spectroscopic voxel was placed over the hand knob region of the right primary motor cortex, based on neuroanatomical guidelines[73] (Fig. 1b). The voxel was shimmed using the automated B0 shimming capabilities of the in-house Visual Display Interface (VDI) libraries (The Weizmann Institute of Science, Israel, www.vdisoftware.net) in MATLAB 2020b (The Mathworks, Natick MA). The MRS acquisition was performed with a SemiLASER (sLASER) sequence (TR/TE = 7000/80 ms, NEX = 36, TA = 4:58 min) previously optimized and validated[74]. Functional MRI data were acquired using a multiband gradient-echo echo-planar imaging sequence. Scanning parameters were implemented according to the Human Connectome Project (HCP) 7T protocol[75] (TR/TE = 1000/22.2 ms, field of view = 208 × 208 mm², matrix size = 130 × 130, voxel-size = 1.6 mm³, 85 slices, multi-band/GRAPPA acceleration factor = 5/2, bandwidth = 1924 Hz/Px, flip angle = 45°). The resting state scans included the acquisition of 420 volumes per scan, while the task paradigms included the acquisition of 710 and 50 volumes on the first and second day, respectively. In addition, spin echo images with opposite phase encoding directions were acquired immediately prior or following each functional acquisition for EPI distortion correction (TR/TE = 3000/60 ms, field of view = 208 × 208 mm², matrix size = 130 × 130, voxel-size = 1.6 mm³, 85 slices, multi-band/GRAPPA acceleration factor = 5/2, bandwidth = 1924 Hz/Px, flip angle = 180°).

**MRS analysis.** MRS pre-processing was carried out using the VDI libraries. Coils were combined via signal-to-noise ratio (SNR) weighting, with weights computed from the reference water and noise scans, using a singular value decomposition algorithm. Spectra were aligned and phase-corrected relative to each other using a previously published robust iterative algorithm[76]. Global zero-order phase-correction was carried out based on the 3.0 ppm creatine peak in the summed spectra. No apodization or zero filling were employed. SPM12 (Wellcome Center for Human Neuroimaging, UCL, UK, http://www.fil.ion.ucl.ac.uk/spm) was used to segment the T1-weighted anatomical images into GM, white matter (WM), and cerebrospinal fluid (CSF) images. Tissue fractions within the spectroscopic voxel were computed using VDI for subsequent use in absolute quantification and as a quality assurance metric. Metabolite quantification was carried out using LCModel[77] version 6.3c, with a basis set containing 17 metabolites: aspartate (Asp), ascorbic acid (Asc), glycerophosphocholine (GPC), phosphocholine (PCh), creatine (Cr), phosphocreatine (PCr), GABA, glucose (Glc), glutamine (Gln), glutamate (Glu), myo-inositol (mI), lactate (Lac), N-acetylaspartate (NAA), N-acetylaspartylglutamate (NAAG), scyllo-inositol (Scyllo), glutathione (GSH), and taurine (Tau), as well as lipids and macromolecules (Lip13a, Lip13b, Lip09, MM09, Lip20, MM20, MM12, MM14, MM17, Lip13a+Lip13b, MM14+Lip13a +Lip13b+MM12, MM09+Lip09, MM20+Lip20). Supplementary Fig. 1 presents representative spectra. Basis functions were simulated by solving the quantum mechanical Liouville equation using VDI, taking into account the full 3D spin profile and the actual pulse waveforms. Absolute quantification was carried out by correcting the metabolite concentrations provided by LCModel for tissue fractions estimated from the segmented images[78], assuming a water concentration of 43.3 M in GM, 35.88 M in white matter (WM) and 5.556 M in cerebrospinal fluid (CSF). Relaxation correction assumed the same value of T2 for GM and WM. We also assumed no metabolites in CSF tissue fractions[79]. The long TR eliminated saturation effects and, consequently, no T1 corrections were required. In addition to the concentration, the relative Cramer Rao Lower Bound (%CRLB) for each metabolite was also obtained.

## fMRI analysis

**Pre-processing.** Functional MRI pre-processing was carried out using the FEAT tool in FSL 6.05 (FMRIB's Software Library, www.fmrib.ox.ac.uk/fsl). The first 5 TRs of the functional data were discarded to allow steady-state magnetization. Registration of the functional data to the high resolution structural images was carried out using boundary based registration algorithm[80]. Registration of high resolution structural to standard space (1 mm MNI152) was carried out using FLIRT[81], and then further refined using FNIRT nonlinear registration. Motion correction of functional data was carried out using MCFLIRT[81], non-brain tissue removal, grand-mean intensity normalization of the entire 4D dataset by a single multiplicative factor, and high-pass temporal filtering was performed with a Gaussian-weighted least-squares straight line fitting with a cut-off period of 100 s. Since we utilized a multivoxel-based pattern analysis (detailed below), minimal spatial smoothing using a Gaussian kernel of 2 mm FWHM was applied to the data[82]. EPI distortion correction of the functional data was carried out with FSL-TOPUP using the acquired spin-echo field maps. In addition, independent component analysis (ICA)-based exploratory data analysis was carried out using FSL's Multivariate Exploratory Linear Decomposition into Independent Components (MELODIC), in order to investigate the possible presence of unexpected artefacts or activations. We implemented the ICA with automatic removal of motion artifacts (ICA-AROMA) tool[83] on the subject-specific spatial ICs and associated time-courses to identify motion-related noise components and denoise the data. The ICA-AROMA denoising strategy identifies ICA noise components based on their location at brain edges and CSF, high frequency content, and correlation with realignment parameters resulting from initial motion correction. ICA-AROMA procedure resulted in a denoised 4D time-course for each participant that was further used in subsequent task and resting-state analyses.

**Task-based analysis.** First-level analysis was carried out on the pre-processed data for each participant. Time-series statistical analysis (pre-whitening) was carried out using FILM with local autocorrelation correction[84]. First-level task regressors were defined based on the onset times of the task blocks and were convolved with a double-gamma hemodynamic response function. Specifically, in the current experiment we have defined two task regressors of interest representing the early-phase and late-phase learning stages in the first day. Regressor of the temporal derivative of the task timing was also included in the analysis. Subject-specific Z-maps were thresholded non-parametrically using clusters determined by $Z > 3.1$ and a corrected cluster significance threshold of $p = 0.05$. Group-level analysis was carried out using FLAME (FMRIB's Local Analysis of Mixed Effects) stage 1. Group-level Z-maps were thresholded non-parametrically using clusters determined by $Z > 3.1$ and a corrected cluster significance threshold of $p = 0.05$. Multiple comparisons correction was spatially restricted to include only GM voxels using a MNI152 GM mask.

**Regions-of-interest definition.** Four functional regions of interest were defined in the current study for the evaluation of M1's functional processing characteristics. First, ROIs of M1 and the right/left putamen were defined as our main MSL-related regions of interest. In addition, to test for the specificity of the selected ROIs in motor memory consolidation, additional ROI of the bilateral posterior cingulate cortex (PCC) was also defined as a control region which is not generally implicated in motor learning[59,61]. M1 ROI was initially defined as the caudal part of the precentral gyrus (the anterior wall of the central sulcus), which directly controls finger movements[85]. This part of the motor cortex was previously shown to be functionally and anatomically differentiated from the more anterior part of the precentral gyrus corresponding to the adjacent dorsal premotor cortex[86]. Then, as the study aim was to investigate aspects of a motor memory formation in M1, we defined the final M1 ROI to be the set of voxels within the caudal

part of the precentral gyrus that demonstrated statistically significant increased activation during both the last (12th) practice block of the first day and the testing block on the second day. This was based on the definition of memory engram cells[87], i.e., activated by a learning experience, and reactivated by subsequent memory reactivation/retrieval, and also previously demonstrated in M1 following motor learning[9]. Importantly, the M1 ROI voxels were located within the MRS voxel. Right and left ROIs of the putamen were defined in the same way. The ROI of the PCC was extracted from the Harvard-Oxford Cortical Atlas, thresholded at 30% and binarized. These ROIs were then used in subsequent analyses.

**Functional processing of M1.** Following pre-processing, the standard space denoised time-courses were used to measure the intra- and inter-regional functional connectivity of M1 using CONN Toolbox v.21a (http://nitrc.org/projects/conn). Resting-state data were further denoised by regressing out the signal of the first component of the CSF[83] extracted with the component-based method (CompCor) implemented in CONN.

**Multivoxel local correlation (MVLC) analysis.** We used the Integrated Local Correlation (ILC) analysis implemented in CONN to construct the MVLC pattern of the voxels within the M1 ROI. The ILC analysis yields the local connectivity of each voxel with its surroundings, which is characterized by the strength and sign of correlation between a given voxel's time-course and the neighboring voxels' time-courses[88]. We used a 1 mm kernel for characterizing the size of the local neighborhoods, in order to express the local correlation of each voxel with its directly surrounding voxels (as the standard space data resolution was 1 mm³). Each voxel's ILC value (i.e., correlation value) was then Fisher Z-transformed. Task-related MVLC pattern was defined as the spatial pattern of ILC values across all M1 ROI voxels during the late phase of learning on the first day. This is based on recent findings suggesting that activity patterns at reactivation correspond to those at the late (but not the early) phase of motor learning in M1[9]. Resting MVLC patterns were defined as the spatial pattern of ILC values across all M1 ROI voxels during either the pre- or post-learning rest periods (in which resting-state BOLD fMRI data were acquired). Next, we measured the similarity between the task MVLC pattern and the pre- and post-learning resting MLVC patterns. This was carried out by transforming the MVLC patterns to vectors and calculating the Pearson correlation between them, i.e., post rest-task correlation and pre rest-task correlation (note that the same elements across vectors correspond to the same M1 ROI voxels). Then, we computed the difference between the two similarity measures (i.e., subtracting the post rest-task similarity and the pre rest-task similarity), resulting in a similarity difference measure for each participant. This in turn enabled us to examine whether post-learning resting MVLC patterns were more similar to the task MVLC patterns compared to the pre-learning rest, therefore presumably reflecting offline memory reactivation[27]).

**Functional connectivity analyses between M1 and other ROIs.** The functional connectivity between the M1 ROI and the putamen/PCC ROIs was calculated using an ROI-to-ROI approach, correlating the average time-courses of the ROIs at rest both before and after the task on the first day, and before the task on the second day. Each correlation value was then Fisher Z-transformed. Differences in functional connectivity were computed by subtracting post and pre values for each participant.

**Structural MRI analysis.** Overnight changes in M1 GM volume were examined using the voxel-based morphometry pipeline implemented in FSL (https://fsl.fmrib.ox.ac.uk/fsl/fslwiki/FSLVBM)[89]. FSL-VBM pre-processing first included non-brain tissue removal using BET, and tissue-type segmentation via a the Automated Segmentation Tool

(FAST) to segment the images into GM, WM, and CSF. FSL FAST also performed bias-field correction for RF/B1-inhomogeneity. Next, the segmented native-space GM images were non-linearly registered with FNIRT to the standard MNI space using the ICBM-152 GM template, in order to create a study-specific GM template. Then, the GM images were non-linearly registered to the study-specific template using FNIRT. Finally, the resulting GM images were modulated by multiplying each voxel in each GM image by the Jacobian of the warp field in order to compensate for the contraction/enlargement due to the non-linear component of the transformation[90]. M1 GM values were extracted from these non-smoothed modulated images by computing the averaged value across all voxels in the M1 ROI. We also extracted the values from the PCC to serve as a control region in further analyses.

**Statistical analyses.** Statistical analyses and visualizations were performed and constructed with MATLAB v21.a, and R v4.1.2. Here, we evaluated post-learning changes (e.g., changes in metabolites concentrations, functional connectivity, behavioral performance) with linear-mixed models using the lme4 package implemented in R. Each mixed effect model in the current study was examined as random intercept and random slope model, enabling the expression of different baseline levels but also difference in the extent of change for each evaluated measure across the participants. To this end, participants ID was used as the random effect and time as the fixed effect. Between-group differences were evaluated by entering an interaction term to the model: dependent variable ~ Time*Group + (Time | ID). Post-hoc pairwise comparisons were corrected for multiple tests using the False Discovery Rate method (FDR)[91]. For the MRS data, the mean concentration and standard deviation of each metabolite, as well as the mean and standard deviation of the CRLB were calculated for each of the time points. Quality assurance of the MRS data followed previously reported metrics: visual inspection for gross artifacts, such as lipid contamination and spurious echoes, metabolites concentrations that were three standard deviations away from the mean of all time-points measurements were excluded from further analyses (following previous recommendation[92] and implementation[74]), as well as water linewidths exceeding 15 Hz FWHM, and SNR of ≤30 from the LCModel output[5,19]. MRS data quality summary is presented in Supplementary Note 1 and Supplementary Table 1. LCModel fitting representation is presented in Supplementary Fig. 1. To evaluate consistency in the MRS voxel placement across the two scanning sessions, repeated measure linear mixed models were used for GM, WM, and CSF tissue fractions of the two voxels. These comparisons did not demonstrate statistically significant tissue fraction differences between the scanning sessions (See Supplementary Note 1). Relationships between continuous variables were assessed using Pearson's correlation coefficient (two-tailed tests). Since we aimed to investigate the unique relationship between Glu or GABA and the other brain/behavior changes, we computed partial correlations while controlling for the other metabolite. The associations between the proposed consolidation and plasticity measures (i.e., functional connectivity metrics and GM volume) and the change in behavioral performance overnight were evaluated using a one-tailed correlation test as directional relationships were hypothesized as part of the rationale of the study. All correlations evaluating a relationship with overnight performance changes were adjusted for performance on the last practice block on the first day, as higher performance on the first day was associated with smaller overnight gains in performance ($p < 0.05$). Corrections for multiple correlations were conducted using FDR within each family of hypothesis tests[38,93]. Comparisons between correlation coefficients (between-groups or within-group) were performed with Fisher's r-to-z transformation using the "cocor" package in R. Within- and between-group correlations comparisons (i.e., between learning and control settings such as the different ROIs or the two groups) were only conducted for statistically significant correlations to reduce the overall number of tests

performed. In addition to excluding Glu or GABA values above or below 3/-3 SD, respectively, outliers for all other continuous measures were defined as values that were more than 3 times the IQR below Q1 or above Q3 and were not included in further analyses.

## Reporting summary
Further information on research design is available in the Nature Portfolio Reporting Summary linked to this article.

## Data availability
Due to medical confidentiality of participants' data the raw imaging data cannot be uploaded to a public repository. The unidentified dataset's spreadsheet generated and analyzed during the current study is available from the corresponding author on request. Source data are provided with this paper.

## Code availability
The MATLAB VDI libraries used to analyze the MRS data are publicly available (https://www.weizmann.ac.il/chembiophys/assaf_tal/software-0). Other MATLAB and R codes used for data processing and visualizations are available from the corresponding author on request.

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

## Acknowledgements

A.T. acknowledges the support of the Monroy-Marks Career Development Fund, the historic generosity of the Harold Perlman Family, and Israeli Science Foundation personal grant 416/20. E.F.H. holds the Calin and Elaine Rovinescu Research Fellow Chair for Brain Research. We would like to acknowledge Edward J. Auerbach, Ph.D. and Małgorzata Marjańska, Ph.D. (Center for Magnetic Resonance Research and Department of Radiology, University of Minnesota, USA) for the development of the pulse sequences for the Siemens platform which were provided by the University of Minnesota under a C2P agreement. We also thank Prof. Nitzan Censor for his thoughtful advice and Dr. Tali Weiss for her assistance with constructing the fMRI task paradigm.

## Author contributions

T.E.—Conceptualization, Methodology, Formal analysis, Investigation, Writing—Original Draft, Writing—Review & Editing, Visualization, Project administration; E.F.H.—Resources, Funding acquisition; A.T.—Conceptualization, Methodology, Writing—Review & Editing, Software, Resources, Supervision, Project administration, Funding acquisition.

## Competing interests

The authors declare no competing interests.
