## [Peer Review File · Nature Communications]

Early excitatory-inhibitory cortical modifications following skill learning are associated with motor memory consolidation and plasticity overnightREVIEWER COMMENTS

Reviewer #1 (Remarks to the Author):

This is a nicely done study that examines how glutamate and GABA levels change on short and long timescales during motor learning. The study design enables examination of the results both immediately after task learning and on a second day. The approach and the analysis appear to be rigorous. There is use of LMM and corrections for multiple comparisons. I think this is a nice addition to the literature. A few things to consider.

1) The title says 'support motor memory consolidation'; this terminology is also present throughout (e.g. line 19). However, really what is shown is a correlation. I think using another term (e.g. predicts or is associated) is more appropriate.

2) There are a lot of comparisons done. In the end it can be a little hard to recall all the associations and the links to short and long timescales (also links to local M1 and cross-area motor cortex-putamen). I think a final figure that summarizes the results will be helpful.

3) I think the follow two papers are relevant and should be discussed in context of the findings here. Boutin ...Doyon J. 2018. Transient synchronization of hippocampo- striato-thalamo-cortical networks during sleep spindle oscillations induces motor memory consolidation and also Lemke et al., Coupling between motor cortex and striatum increases during sleep over long- term skill learning, eLife 2021.

Reviewer #2 (Remarks to the Author):

The manuscript by Eisenstein and co-authors employs a multimodal approach for the investigation of neurochemical, functional connectivity and structural changes associated with motor learning process. In particular, the authors are interested in how are the dynamics of glutamate and GABA linked to the process of motor memory consolidation and to changes in functional connectivity and to structural plasticity in relevant brain areas.

The study is highly innovative, very well designed and performed at all levels - from the data acquisition to the data analysis. Particularly noteworthy is the attention to studying the different temporal dynamics of excitation and inhibition, opening the way to fresh interpretation of the role of both in motor learning.

As such, it is an excellent example of an important neuroscientific topic investigated with the best tools available.

Following are some comments that I hope the authors consider useful.

1. A general comment regarding statistical significance: there is no such accepted term as "statistical trend". Blurring the lines between statistical significance and lack thereof does not do good service to science, nor to this work. Here's an opinion from researchers in another field: <https://www.ncbi.nlm.nih.gov/pmc/articles/PMC6440716/> I suggest sticking to the standard conventions of statistical significance, and focus on the significant findings of this work. Also, it is not clear whether any multiple comparison correction has been applied to the overall rather large number of statistical tests here. A comment on that would be welcome.
2. A central debate in the neurochemistry/MR spectroscopy community relates to the nature of changes in Glu concentration upon brain activation. Glu serves in the dual role of excitatory neurotransmitter and intermediate product of the TCA cycle, hence its concentration changes may reflect neurotransmission and metabolism. This is in stark contrast to GABA, where changes in concentration can be more confidently attributed to inhibitory neurotransmission. It would be beneficial if the authors consider how to include this in their interpretation of observed Glu changes. The authors offer an interesting, but to my knowledge unsubstantiated hypothesis regarding increase in cytosolic Glu as release from vesicular glu. I'm not aware of any experimental evidence for this hypothesis, and I'm not sure this should be even be given as an explanation.
3. I find it difficult to understand some of the results related to neurotransmitter dynamics, for example the one shown in Figure 2, panel C (post-learning neurochemical dynamics depend on baseline metabolite levels): The results are crisp and convincing, but I don't see how can a learning process (unless there's a 50% chance of learning and forgetting...) results in as many positive changes as negative changes. In other words - concentration change of 0% for both GABA and Glu falls directly in the middle of the data shown. I'll be thankful for an explanation/interpretation. Also, in figure 3B and C there's an outlier in all graphs. I wonder whether this outlier was included in the analysis.
4. Small technical comment: P8 line 184 mentions "spin echo field maps with opposite phase encoding directions". Typically it's either distortions are corrected with field maps (then no need for opposite polarities of the PE gradients, and they are acquired with gradient echo and not spin echo), OR a method that does not use field maps but acquisition of EPI (spin echo or gradient echo, depending on the experiment) images are taken and are the basis for the distortion correction. Please check and correct accordingly.

Reviewer #3 (Remarks to the Author):

Review > Early excitatory-inhibitory modifications in the motor cortex following skill learning support motor memory consolidation and plasticity overnight

Einsenstein and colleagues performed a very instructive behavioral and multimodal neuroimaging study on the excitatory-inhibitory mechanisms underlying the consolidation of motor memories. They conducted their study using BOLD fMRI, resting –state fMRI and MRS at 7T within the M1 and the putamen of human volunteers to examine the short and longer time responses prior to and following a motor-learning task as well as following a night of sleep. One of the main findings of their study is that excitatory (Glu) and inhibitory (GABA) changes supported different processes following learning-induced structural, functional and behavioral overnight modifications in M1. Moreover both Glu and GABA changes supported learning-induced consolidation mechanisms.

The study of Einsenstein and colleagues highlights some potentially fundamental processes of memory consolidation within the human brain and in particular the differing but connected and fundamental roles of Glu and GABA. The mechanisms of memory consolidation and the roles of important neurotransmitters such as Glu and GABA remain poorly understood and the present study represents a step forward for a better understanding of the underlying neurochemical mechanisms of memory consolidation. This study also demonstrates the potential of multimodal neuroimaging studies and notably of investigations with MRS at higher field strengths.

The study was well-conducted and appears robust. I recognize here the thoroughness and consistency of the neuroimaging studies performed by Dr Tal Assaf. Nevertheless, I have a few comments and questions that need to be addressed by the authors.

Abstract

Introduction:

- Just to add some precision, it might be good to indicate that the authors are talking about Glu and GABA concentrations and not about signals, which might be confusing for “non-MRS” readers.

P4 L 68: Magnetic resonance spectroscopy (MRS) is currently the only method capable of non-invasively quantify excitation and inhibition in the human brain, by directly quantifying the concentrations of Glu and GABA.

- Change to quantifying

- Authors could add a little bit more explanations on the role of the putamen in motor learning

- Why did authors decide not to add in a seed within the hippocampus? Is it because of the different structures that might have differing implications ? Please comment on this point

- Materials and Methods

What was the CRLB threshold for reliability ?

The MRS quality threshold in suppl. Materials is a very interesting addition. It would be nice to also add representative examples of the different spectra acquired as a function of time.

In Table S1, are CRLBs reported as % and linewidths in Hz? Please add units

Authors did not report or talk about the macromolecular spectrum. Was it also simulated ? It is not shown in Figure S1

Also, the Glu and GABA concentration changes are reported as a % change in the overall paper. Could authors also add a table of with absolute concentrations across the different measurements.

Were there other metabolite concentration changes? NAA or ml for example?

Authors decided to use a semi-Laser sequence at a TE=80 ms for the measurement of GABA and Glu concentrations since they showed in an earlier paper (ref 62) that the reproducibility of measurements were improved compared to using MEGA-semi-LASER. However, earlier measurements were performed in a different structure (dACC). How reliable and reproducible were GABA concentration measurements in M1 which is usually more difficult to shim than dACC? Could authors provide some data (CVs, test-retest...) showing evidence that GABA and Glu had similar reproducibility in M1 than in dACC with the non-edited sequence they used?

Statistical analysis

- Although Pearson's correlation does not assume normality, it would be interesting to test for the normality of the different distributions. Since basically anything can correlate with anything and although the number of points is sufficient for an appropriate Pearson's testing, it would be interesting that authors justify their choice of using this test.

Results

- I don't think that the evolution of Glutamate concentrations shown in Figure 2A can really be considered significant with p-values >0.05 This is an eternal debate of whether p=0.075 can be considered nearly significant. Figure 2A does not show a clear trend.

- However the correlations between Glu and GABA concentrations are very convincing. Interestingly GABA concentrations remained in the same range. At 30 min post-learning, the distribution of points seems more skewed towards lower Glu concentrations compared to previous points.

- Pre and post-learning negative correlations are also convincing. Could authors comment on post-learning Glu and GABA levels above 20 and 50% respectively. Is this plausible?

- Figure 3B shows that many participants had decreased performances also corresponding to lower Glu or GABA levels. Did authors consider grouping patients responding positively and those responding negatively? Could these responses be a result of adaptation or habituation_?
- In figure 3C, the correlation between performance change and GABA concentration is not significant and the trend to correlation mentioned in the text can be questioned
- Figure 4B The trend for GABA is again not convincing
- P20 L 435: Same as before I don't know what is a statistical trend
- In Figure 6, the title "decreased GABAergic inhibition" is not appropriate. GABA can also be excitatory ... Authors should re-phrase
- Why did authors decide not to correlate BOLD-fMRI metrics with Glu or GABA? Please comment?

Discussion and conclusion

The discussion and conclusion are well written and address several interesting issues using literature findings in animal models that can afford higher spatial resolutions. However, I find the discussion very general and therefore very speculative about the results obtained. I would like that authors discuss their results with a bit more specificity. It would be nice to propose methods to further validate their findings. Unfortunately although I find this study relevant and well-conducted, I find the GABA results/correlations not always convincing at this stage. Could it be related to the GABA measurement reliability and reproducibility in M1?

I would expect the authors to discuss with more details the changes occurring at 30- minutes post-learning. While authors mentioned LTP processes, they do not really discuss the timing of the effects. Moreover, adaptation and habituation are not discussed. Other mechanisms could also be mentioned and could be discussed such as cortical neurogenesis or cell swelling as well sleep/awake states which could have influenced neurochemistry. Some explanations may also be found/discussed regarding specific Glutamatergic and GABAergic projections from/to M1.

Reviewer #4 (Remarks to the Author):

This is an extremely exciting manuscript that seeks to combine MRS, a methodological strength, with other neuroimaging in the context of a motor learning paradigm. The repeat-measures/longitudinal design is a real strength and the findings are intriguing. This paper describes important work that will be of interest to a broad spectrum of scientists, from neuroscientists to imaging scientists to neurochemists and motor neurologists. Where the results perhaps fall short of being fully compelling (and where the paper would be improved with additional experiments) is in the degree of controls adopted.

1. Additional controls are needed to demonstrate effects. For example, in Figure 2A there is an increase in Glu around the motor learning paradigm. However, there is not control arm demonstrating that there is no increase in control regions of the brain, or in subjects not engaged in a motor learning task. It

would also be important to show (in Supplementary) the same plots for major metabolites (e.g. NAA, Cr?) and Glu-like minor metabolites (Gln, ml ?), to address the extent to which the effects seen are Glu-specific.

2. In general, while the sample size is relatively good for a multi-scan MRS study, there are so many variables in play (pre, post, changes, metabolites etc) that it is difficult to know how strongly the stated hypotheses were fixed before the study began. For publication in a journal like NC, I would expect to see a full replication (perhaps powered for one-tailed statistics).

I wrote this comment even before I realized the full range of neuroimaging performed (including resting-state MRI, which has almost infinite potential metrics). I appreciate that replication is a high bar, but that is what I think is required here.

3. This goes especially for results relating neurochemical change to structural change. Where there is no global change between timepoints in either axis and a correlation where some people reduce e.g. GABA and reduce tissue volume and others increase both, there is real concern about the independence of measures. The tissue-corrected GABA results and the structural measure presumably make use of the same T1 images. In replicating this result, it would be important to repeat the T1, once for tissue correction and once for volume measures. Otherwise the variables correlated are not statistically independent and noise in the segmentation will contribute (perhaps lightly) to noise in the metabolite measures. An alternative way to show this is a non-issue is to plot the same results without tissue correction or using a Cr-referenced measure.

4. That a short learning paradigm results in a meaningful structural changes detectable by structural imaging is a challenging idea... that such a change is positive in some subjects and negative in others and still is a meaningful response to the training is even harder. What is the proposed reason for this range?

REVIEWER COMMENTS

We thank the Reviewers for their insightful comments and have made corresponding substantive changes to the manuscript. Some of the major revisions include:

- The recruitment of an additional group of “control” participants (n=21) who did not perform a motor learning task to serve as a resting control group. Instead of performing the MSL task, those participants laid with their eyes open (with fixation) during an equivalent period of time and were also scanned before and after according to the full experimental protocol. This group confirms that the effects seen for GABA and Glu in motor memory consolidation and overnight plasticity are unique to the group undergoing learning.
- Additional within-group control analyses showing that the same patterns of functional and structural changes do not take place in control regions (i.e., PCC), further lending credibility to our findings. Furthermore, changes in other metabolites such as NAA and Gln were not evident following learning.
- The discussion and references have been expanded considerably following the Reviewers comments to better interpret and support the findings of the current study.

Reviewer #1 (Remarks to the Author):

This is a nicely done study that examines how glutamate and GABA levels change on short and long timescales during motor learning. The study design enables examination of the results both immediately after task learning and on a second day. The approach and the analysis appear to be rigorous. There is use of LMM and corrections for multiple comparisons. I think this is a nice addition to the literature. A few things to consider.

R1.1 The title says ‘support motor memory consolidation’; this terminology is also present throughout (e.g. line 19). However, really what is shown is a correlation. I think using another term (e.g. predicts or is associated) is more appropriate.

We thank the Reviewer for her/his comment – we have edited the title and text accordingly (p.1 line 17; p.16, line 416; p.18 line 448; p.25 line 604; p.30, line 745).

R1.2 There are a lot of comparisons done. In the end it can be a little hard to recall all the associations and the links to short and long timescales (also links to local M1 and cross-area motor cortex-putamen). I think a final figure that summarizes the results will be helpful.

We thank the Reviewer for her/his suggestion - we have now added a summarizing figure of all significant associations observed between Glu or GABA changes and neuro-behavioral correlates of motor memory consolidation to the Supplementary Materials (please see Figure S4, and the referencing to it at the beginning of the Discussion in p.24, line 550-551).

R1.3 I think the follow two papers are relevant and should be discussed in context of the findings here. Boutin ...Doyon J. 2018. Transient synchronization of hippocampo- striato-thalamo-cortical networks during sleep spindle oscillations induces motor memory consolidation and also Lemke et al., Coupling between motor cortex and striatum increases during sleep over long- term skill learning, eLife 2021.

We have added the following discussion on these papers to the manuscript (pp.29-30, lines 724-744):

“While it is now generally accepted that post-learning sleep plays a role in motor memory consolidation following explicit skill learning²³, two previous studies in both rats²² and humans⁸¹ provided evidence

that learning-related modulation of cortico-striatal activity take place during sleep, and that particularly non-rapid eye movement (NREM) sleep and sleep spindles are essential in shaping this connection. Lemke and colleagues (2021)²² demonstrated that increases in functional connectivity between neurons in M1 and in the dorsolateral striatum (generally homologues to the putamen in humans²⁵) occurred offline following learning, and not during learning itself as measured immediately after practice. Boutin and colleagues (2021)⁸¹ further found using electroencephalography (EEG) that greater spindle band coherence between M1 and the putamen in the contralateral hemisphere to the training hand was related to overnight improvements in skill performance. Our results may complement those findings as we found short-term glutamatergic changes in M1 to associate with overnight increases in the functional connectivity between M1 and the putamen, but not shortly after the learning session. Moreover, this relationship was specific to the contralateral hemisphere to the learning hand (i.e., right M1 and putamen), which by itself was related to greater improvements in performance, regardless of neurochemical changes following learning (although did not survive multiple comparisons correction). Furthermore, while Lemke and colleagues (2021)²² found that offline increases in cortico-striatal connectivity correlated with skill performance, they also reported that only 35% of M1 and striatal electrode pairs that were examined demonstrated offline increases in connectivity, with the others electrode pairs exhibiting either decreases or no change. This may support our results as we did not find significant overnight changes in M1-putamen connectivity at the group-level following learning.”

Reviewer #2 (Remarks to the Author):

The manuscript by Eisenstein and co-authors employs a multimodal approach for the investigation of neurochemical, functional connectivity and structural changes associated with motor learning process. In particular, the authors are interested in how are the dynamics of glutamate and GABA linked to the process of motor memory consolidation and to changes in functional connectivity and to structural plasticity in relevant brain areas.

The study is highly innovative, very well designed and performed at all levels - from the data acquisition to the data analysis. Particularly noteworthy is the attention to studying the different temporal dynamics of excitation and inhibition, opening the way to fresh interpretation of the role of both in motor learning. As such, it is an excellent example of an important neuroscientific topic investigated with the best tools available. Following are some comments that I hope the authors consider useful.

R2.1 A general comment regarding statistical significance: there is no such accepted term as "statistical trend". Blurring the lines between statistical significance and lack thereof does not do good service to science, nor to this work. Here's an opinion from researchers in another field: <https://www.ncbi.nlm.nih.gov/pmc/articles/PMC6440716/> I suggest sticking to the standard conventions of statistical significance, and focus on the significant findings of this work. Also, it is not clear whether any multiple comparison correction has been applied to the overall rather large number of statistical tests here. A comment on that would be welcome.

We thank the Reviewer for this comment – we have now discarded any reference to “statistical trend” from the text.

Furthermore, FDR correction for multiple comparisons have been implemented in this manuscript – we now emphasize it in the Methods section (p.11, line 331 and p.12 line 353)

R2.2. A central debate in the neurochemistry/MR spectroscopy community relates to the nature of changes in Glu concentration upon brain activation. Glu serves in the dual role of excitatory neurotransmitter and intermediate product of the TCA cycle, hence its concentration changes may

reflect neurotransmission and metabolism. This is in stark contrast to GABA, where changes in concentration can be more confidently attributed to inhibitory neurotransmission. It would be beneficial if the authors consider how to include this in their interpretation of observed Glu changes. The authors offer an interesting, but to my knowledge unsubstantiated hypothesis regarding increase in cytosolic Glu as release from vesicular glu. I'm not aware of any experimental evidence for this hypothesis, and I'm not sure this should be even be given as an explanation.

We thank the Reviewer for raising this meaningful consideration. Regarding the discrepancy between neurotransmission and metabolism, although there is no way to truly resolve this issue using the current application of ^1H -MRS, we have added the following paragraph to the text (pp.26-27, lines 637-651):

"In addition, it should be noted that ^1H -MRS does not enable to distinguish between intra-synaptic, extra-synaptic or intracellular compartments, or between different cell types. Therefore, while Glu subserves physiological functions in both neurons and astrocytes and plays a dual role in excitatory neurotransmission and cellular energy metabolism, the origin and physiological pathways of the MRS Glu signal are currently could not be discernible using ^1H -MRS. Therefore, Glu concentration changes may also reflect increased anaplerotic reactions subserving the tricarboxylic acid cycle (TCA) cycle during or following increased neural activity. Furthermore, there is a well-documented coupling between oxidative metabolism (and the potential energetic role of Glu) and the glutamate-glutamine cycling⁷⁶. While a compartmental shift mechanism based on the visibility of cellular compartments to MRS may support the contribution of increased vesicular release and changes in neurotransmission to the dynamic changes observed in Glu, it needs further investigation. Nevertheless, although we cannot point out to the physiological origin of the increased Glu associated with motor memory consolidation in the current study, either increased neurotransmission or increased metabolic activity could reflect elevated levels of neural processing following learning that are bound to the early stages of offline consolidation processes."

Regarding the hypothesis of increased cytosolic Glu - please note that we have provided references in the text (p.26, line 621-626) that may support the possibility of increased transmitter release following learning or intense activation:

Markram H, Tsodyks M. Redistribution of synaptic efficacy between neocortical pyramidal neurons. *Nature*. 1996;382(6594):807-810. doi:10.1038/382807a0

Eder M, Zieglgänsberger W, Dodt H-U. Neocortical long-term potentiation and long-term depression: site of expression investigated by infrared-guided laser stimulation. *J Neurosci Off J Soc Neurosci*. 2002;22(17):7558-7568. doi:10.1523/JNEUROSCI.22-17-07558.2002

Zakharenko SS, Zablow L, Siegelbaum SA. Visualization of changes in presynaptic function during long-term synaptic plasticity. *Nat Neurosci*. 2001;4(7):711-717. doi:10.1038/89498

Castillo PE. Presynaptic LTP and LTD of excitatory and inhibitory synapses. *Cold Spring Harb Perspect Biol*. 2012;4(2). doi:10.1101/cshperspect.a005728

Kida H, Tsuda Y, Ito N, et al. Motor Training Promotes Both Synaptic and Intrinsic Plasticity of Layer II/III Pyramidal Neurons in the Primary Motor Cortex. *Cereb Cortex*. 2016;26(8):3494-3507. doi:10.1093/cercor/bhw134

R2.3 I find it difficult to understand some of the results related to neurotransmitter dynamics, for example the one shown in Figure 2, panel C (post-learning neurochemical dynamics depend on baseline metabolite levels): The results are crisp and convincing, but I don't see how can a learning process (unless there's a 50% chance of learning and forgetting...) results in as many positive changes as negative changes. In other words - concentration change of 0% for both GABA and Glu falls directly in the middle of the data shown. I'll be thankful for an explanation/intepretation. Also, in figure 3B and C there's an outlier in all graphs. I wonder whether this outlier was included in the analysis.

We thank the Reviewer for this comment. To clarify, Figure 2, panel C does not present the learning process, only the neurochemical changes following the learning or the resting control condition. Therefore, it does not imply anything on the learning process in terms of behavioral learning or forgetting (please note that the learning process resulted in significant behavioral improvements, both during the first day of initial learning and overnight as can be appreciated in Figure 3A (p.17).

Regarding the observation that “concentration change of 0% for both GABA and Glu falls directly in the middle of the data shown” – this is the result/expression of the linear relationship between those variables and would not be the case if the variables were not linearly correlated. Also, this is the result of the fact that there were no group-level changes following learning. Please note that a similar relationship between pre and post-learning metabolic changes was also observed in previous works. We have now added the following paragraph to the Discussion referring to this observation (p.25, lines 591-602):

“We also found that the extent and direction of GABAergic changes over time were strongly related to baseline GABA levels. Specifically, higher GABA levels at baseline were associated with greater reduction in GABA following either learning or the resting condition, and vice versa. This inverse relationship is in accordance with previous findings⁵³, and is therefore suggestive of a general homeostatic mechanism that act to regulate GABA levels over time, as was also proposed to occur during sleep following learning⁶¹. In contrast, we found a similar inverse relationship for Glu levels following learning, but not following rest. This in turn, may suggest that short-term Glu dynamics, in contrast to GABA, may be more sensitive and reflective of recent neural responses to external stimuli. However, while it is possible that this inverse relationship also reflects a homeostatic mechanism following activation (i.e., the learning task), we did not find an association between the extent of finger movements or M1 activation during the MSL task and the extent of post-learning changes in either Glu or GABA.”

Further, we have now removed the outlier from figure 3B-C (p.17). Please note that it did not change the nature of the relationships presented in the plots (significant relationships remained statistically significant).

R2.4 Small technical comment: P8 line 184 mentions "spin echo field maps with opposite phase encoding directions". Typically it's either distortions are corrected with field maps (then no need for opposite polarities of the PE gradients, and they are acquired with gradient echo and not spin echo), OR a method that does not use field maps but acquisition of EPI (spin echo or gradient echo, depending on the experiment) images are taken and are the basis for the distortion correction. Please check and correct accordingly.

We thank the Reviewer for directing our attention to this issue – we have rephrased it in the text (p.7, lines 188-189).

Reviewer #3 (Remarks to the Author):

Einsenstein and colleagues performed a very instructive behavioral and multimodal neuroimaging study on the excitatory-inhibitory mechanisms underlying the consolidation of motor memories. They conducted their study using BOLD fMRI, resting –state fMRI and MRS at 7T within the M1 and the putamen of human volunteers to examine the short and longer time responses prior to and following a motor-learning task as well as following a night of sleep. One of the main findings of their study is that excitatory (Glu) and inhibitory (GABA) changes supported different processes following learning-induced structural, functional and behavioral overnight modifications in M1. Moreover both Glu and GABA changes supported learning-induced consolidation mechanisms.

The study of Einsenstein and colleagues highlights some potentially fundamental processes of memory consolidation within the human brain and in particular the differing but connected and fundamental roles of Glu and GABA. The mechanisms of memory consolidation and the roles of important neurotransmitters such as Glu and GABA remain poorly understood and the present study represents a step forward for a better understanding of the underlying neurochemical mechanisms of memory consolidation. This study also demonstrates the potential of multimodal neuroimaging studies and notably of investigations with MRS at higher field strengths. The study was well-conducted and appears robust. I recognize here the thoroughness and consistency of the neuroimaging studies performed by Dr Tal Assaf. Nevertheless, I have a few comments and questions that need to be addressed by the authors.

R3.1 [Introduction] Just to add some precision, it might be good to indicate that the authors are talking about Glu and GABA concentrations and not about signals, which might be confusing for “non-MRS” readers.

We thank to Reviewer for this comment – Our results are based on absolute quantification, as detailed in the Methods section, and we have now added this clarification throughout the Introduction (p.4, lines 82 & 85, p.5 lines 104 & 108)

R3.2 [Introduction] P4 L 68: Magnetic resonance spectroscopy (MRS) is currently the only method capable of non-invasively quantify excitation and inhibition in the human brain, by directly quantifying the concentrations of Glu and GABA – Change to “quantifying”

We have now edited it in the text (p.3, line 63).

R3.3 [Introduction] Authors could add a little bit more explanations on the role of the putamen in motor learning

We have updated the Introduction as follows (p.3, lines 77-87):

“The consolidation of new motor skill memories has been shown to depend on cortico-striatal brain circuits⁸. The motor cortex serves as a major source of input to the sensorimotor striatum²⁰. Cortico-striatal coupling and plasticity over training sessions are essential for the refinement of skilled behaviour^{8,9,21}, and increased coupling between the motor cortex and the striatum following motor skill learning has been shown to occur offline²². Moreover, motor cortical-striatal interactions have been suggested to be especially important for the consolidation of the egocentric/movement representation component during motor sequence learning²³. Previous studies have suggested that striatal structures are differentially engaged at different levels of the learning process, namely that the associative striatum is more involved during the initial stages of skill learning and acquisition, while the sensorimotor striatum which receives inputs from the motor cortex²⁴ and generally corresponds in primates to the putamen²⁵, is linked with consolidation and gradual acquisition of behavior^{25,26}.”

R3.4 [Introduction] Why did authors decide not to add in a seed within the hippocampus? Is it because of the different structures that might have differing implications? Please comment on this point.

The role of the hippocampus in motor memory consolidation has indeed started to gain increasing attention in recent years. However, as now emphasized in the Introduction (p.3, lines 77-87), M1 and the striatum are tightly operating following motor skill learning to consolidate the motoric/movement representation of the acquired skill. The hippocampus on the other hand is more related to the spatial/allocentric representation of the skill. Given the already-substantial size of the manuscript and the many results included, we aimed to keep the research questions/outcomes presented in this study as focused as possible.

R3.5 [Materials and Methods] What was the CRLB threshold for reliability?

As stated in the Results section, we did not use a CRLB criteria for reliability, as was previously recommended (Kreis 2016) and implemented (Finkelman et al. 2022), and excluded metabolite measurements that were ± 3 SD from the overall mean. Please note that in addition to performing the study on a 7T scanner which enables a more reliable quantification of low levels metabolites such as GABA, as well as removing low or high potential outliers/unreliable concentrations (above or below ± 3 SD), we also used exclusion criteria for the MRS data that were based on the overall quality of each acquired spectrum (e.g., SNR and water linewidth) as also implemented before (Finkelman et al. 2022; Kolasinski et al. 2019; Maruyama et al. 2021).

Finkelman, Tal, Edna Furman-Haran, Rony Paz, and Assaf Tal. 2022. "Quantifying the Excitatory-Inhibitory Balance: A Comparison of SemiLASER and MEGA-SemiLASER for Simultaneously Measuring GABA and Glutamate at 7T." NeuroImage 247:118810. doi: <https://doi.org/10.1016/j.neuroimage.2021.118810>.

Kolasinski, James, Emily L. Hinson, Amir P. Divanbeighi Zand, Assen Rizov, Uzay E. Emir, and Charlotte J. Stagg. 2019. "The Dynamics of Cortical GABA in Human Motor Learning." The Journal of Physiology 597(1):271–82. doi: <https://doi.org/10.1113/JP276626>.

Kreis, Roland. 2016. "The Trouble with Quality Filtering Based on Relative Cramér-Rao Lower Bounds." Magnetic Resonance in Medicine 75(1):15–18. doi: [10.1002/mrm.25568](https://doi.org/10.1002/mrm.25568).

Maruyama, S., M. Fukunaga, S. K. Sugawara, Y. H. Hamano, T. Yamamoto, and N. Sadato. 2021. "Cognitive Control Affects Motor Learning through Local Variations in GABA within the Primary Motor Cortex." Scientific Reports 11(1):18566. doi: [10.1038/s41598-021-97974-1](https://doi.org/10.1038/s41598-021-97974-1)

R3.6 [Materials and Methods] The MRS quality threshold in suppl. Materials is a very interesting addition. It would be nice to also add representative examples of the different spectra acquired as a function of time.

This has now been added to supplementary Figure 1.

R3.7 [Materials and Methods] In Table S1, are CRLBs reported as % and linewidths in Hz? Please add units

Yes, we have now added this information to Table S1.

R3.8 [Materials and Methods] Authors did not report or talk about the macromolecular spectrum. Was it also simulated? It is not shown in Figure S1

Macromolecules were also simulated and we have now added this clarification to the Methods (p.8, line 207-210). Please note that Figure S1 does present MM basis functions, specifically in this spectrum MM09, MM20, MM12, MM14, MM17.

R3.9 [Materials and Methods] Also, the Glu and GABA concentration changes are reported as a % change in the overall paper. Could authors also add a table of with absolute concentrations across the different measurements.

We have now added this to the Supplementary Materials - please see Table S2.

R3.10 [Materials and Methods] Were there other metabolite concentration changes? NAA or ml for example?

Following the Reviewer's and another Reviewer's comments we examined whether there were significant changes in other major or Glu-like minor metabolites, i.e., NAA and Gln, respectively, following learning. No significant changes were observed in NAA ($p=.119$) or Gln ($p=.602$) following learning (see **Supplementary Figure 2** and the Results section p.13, line 378-380).

R3.11 [Materials and Methods] Authors decided to use a semi-Laser sequence at a TE=80 ms for the measurement of GABA and Glu concentrations since they showed in an earlier paper (ref 62) that the reproducibility of measurements were improved compared to using MEGA-semi-LASER. However, earlier measurements were performed in a different structure (dACC). How reliable and reproducible were GABA concentration measurements in M1 which is usually more difficult to shim than dACC? Could authors provide some data (CVs, test-retest...) showing evidence that GABA and Glu had similar reproducibility in M1 than in dACC with the non-edited sequence they used?

The intra-subject test-retest reproducibility as measured with CV across all participants and MRS runs was 5.06% for glutamate and 21.96% for GABA, and follow the CVs reported in (Finkelman et al, 2022). Also, please note that other quality measurements (i.e., water linewidth) are even numerically better than the ones reported in (Finkelman et al, 2022). Also, the SNR of the MRS runs in the current study is numerically higher than previously reported in M1 on 7T scanners (Kolasinski et al. 2019; Maruyama et al. 2021).

R3.12 [Statistical Analysis] Although Pearson's correlation does not assume normality, it would be interesting to test for the normality of the different distributions. Since basically anything can correlate with anything and although the number of points is sufficient for an appropriate Pearson's testing, it would be interesting that authors justify their choice of using this test.

As the Reviewer stated the Pearson's correlation does not assume normality, however, we have now added the distribution of variables in all correlations' scatter plots throughout the manuscript as requested.

R3.13 [Results] I don't think that the evolution of Glutamate concentrations shown in Figure 2A can really be considered significant with pvalues >0.05 This is an eternal debate of whether $p=0.075$ can be considered nearly significant. Figure 2A does not show a clear trend.

We have now removed it from the figure (p.15).

R3.14 [Results] However the correlations between Glu and GABA concentrations are very convincing. Interestingly GABA concentrations remained in the same range. At 30 min post-learning, the distribution of points seems more skewed towards lower Glu concentrations compared to previous points.

This seemingly skewed pattern is due to the highest Glu point in the plot, which we have now removed from the data due to a request from another Reviewer (p.15). The majority of the data are distributed between 7.0 to 9.0 mM for Glu in each time point.

R3.15 [Results] Pre and post-learning negative correlations are also convincing. Could authors comment on post-learning Glu and GABA levels above 20 and 50% respectively. Is this plausible?

All Glu changes were less than 20% as can be appreciated from Figure 2C (p.15) – highest value was 18.5% change.

While there are three GABA change values in the learning group that are above 50%, those values are derived from high quality MRS measurements (based on the MRS quality measures inclusion/exclusion criteria and outliers' removal). Also, given that the data were acquired using a 7T scanner, higher range of GABA levels (and therefore also changes) can be observed in the study due to the higher SNR and detection power, and therefore included in the data analysis. Furthermore, given that the test-retest for GABA was ~22%, a 50% change is not significantly uncommon. Assuming a normal distribution, then ~7.5% of total results should exhibit a 50% change.

Please also note that a similar range of physiological changes in GABA concentrations was also previously reported in M1 following non-invasive brain stimulation, supporting the biological plausibility of such changes:

1. Kim S, Stephenson MC, Morris PG, Jackson SR. *tDCS-induced alterations in GABA concentration within primary motor cortex predict motor learning and motor memory: a 7 T magnetic resonance spectroscopy study.* *Neuroimage.* 2014 Oct 1;99:237-43. doi: 10.1016/j.neuroimage.2014.05.070. Epub 2014 Jun 3. PMID: 24904994; PMCID: PMC4121086.

Lastly, please note that removing those three values above 50% change (see plot below) does not significantly change the correlation strength between baseline GABA levels and post-learning GABA changes ($r=-.71, p<.001$ vs. $r=-.68, p<.001$), and therefore we have decided to include those values in the analysis.

R3.16 [Results] Figure 3B shows that many participants had decreased performances also

corresponding to lower Glu or GABA levels. Did authors consider grouping patients responding positively and those responding negatively? Could these response be a result of adaption or habituation_?

Figure 3B shows partial correlation plots (as described in the Methods section – p.12, lines 350-352) and therefore both axes present mean-centred/demeaned residualized values derived from the partial correlation analysis. In fact, most participants (n=27) demonstrated some degree of improvement in performance overnight (as expected in this kind of a learning task). Only 3 participants demonstrated decrease in performance overnight, while 6 additional participants demonstrated levelled performance overnight. Therefore, adaption or habituation were probably not evident in the current experiment.

We have now clarified this discrepancy in the Figure's axes (p.17) and in the Results section (p.16, lines 420-422).

R3.17 [Results] In figure3C, the correlation between performance change and GABA concentration is not significant and the trend to correlation mentioned in the text can be questioned

We have now rephrased the text (p.16, lines 426-428)

R3.18 [Results] Figure4B The trend for GABA is again not convincing

We have now edited the Figure and significance levels were highlighted with bold for significant values only (p.23). Please note that we do not make any subjective judgment in the figure, only presenting the correlation strength and p-values. Please also note that we have moved this figure to the last subtopic of the Results section, and it is now Figure 6 and not 4.

R3.19 [Results] P20 L 435: Same as before I don't know what is a statistical trend

All references to "statistical trends" have now been eliminated from the text.

R3.20 [Results] In Figure 6, the title "decreased GABAergic inhibition" is not appropriate. GABA can also be excitatory ... Authors should re-phrase

We have rephrased the title of the Figure (which is now Figure 5) accordingly (p.22)

R3.21 [Results] Why did authors decide not to correlate BOLD-fMRI metrics with Glu or GABA? Please comment?

We have now added an analysis of the relationship between Glu/GABA changes following learning and BOLD activation level during the MSL task (pp.13-14, lines 397-405). We have not delved deeper into other aspects of BOLD since, as with our response to the comment on the hippocampus, we wanted to stay focused on our predefined questions of interest due to the large extent of data already presented in this manuscript.

R3.22 [Discussion] The discussion and conclusion are well written and address several interesting issues using literature findings in animal models that can afford higher spatial resolutions. However, I find the discussion very general and therefore very speculative about the results obtained. I would like that authors discuss their results with a bit more specificity. It would be nice to propose methods to further validate their findings.

The discussion have now been expanded over its different subtopics (e.g., behavioral, functional, and structural findings) to better interpret and support the findings of the current study (p.24, line 544).

R3.22 [Discussion] Unfortunately although I find this study relevant and well-conducted, I find the GABA results/correlations not always convincing at this stage. Could it be related to the GABA measurement reliability and reproducibility in M1?

First, note that despite the correlation between GABA changes following 30 minutes and overnight changes in M1 volume, we did not observe other significant correlations between GABA and any other neural or behavioral consolidation-related measure.

However, given that the neuroimaging methodology is sound, and statistical analyses is fairly standard and conclusive, we are unsure why this (single) effect remains unconvincing:

1. The acquisition sequence was thoroughly validated in a previous publication, and the test-retest for GABA and Glu are even superior in the current study (see R3.15).
2. Strict exclusion criteria were uniformly stated and applied based on linewidths and other spectral quality assurance metrics.
3. SNR and overall data quality, as assessed visually from the spectra, were uniformly excellent.
4. False discovery rate corrections were performed throughout.
5. **New:** A newly-added control group does not exhibit the significant correlation with GABA change shown in the learning group.
6. **New:** A newly-added control ROI (i.e., PCC) does not exhibit the significant correlation with GABA change shown in the learning group.

It might not be a *large* effect, but we see no reason to doubt our statistically significant results (which also make sense from a neurobiological standpoint), and no methodological issue to warrant doing so. If there is any such specific issue the Reviewer wishes to raise, we will gladly do our best to address it.

R3.23 [Discussion] I would expect the authors to discuss with more details the changes occurring at 30- minutes post-learning. While authors mentioned LTP processes, they do not really discuss the timing of the effects. Moreover, adaption and habituation are not discussed. Other mechanisms could also be mentioned and could be discussed such as cortical neurogenesis or cell swelling as well sleep/awake states which could have influenced neurochemistry. Some explanations may also be found/discussed regarding specific Glutamatergic and GABAergic projections from/to M1.

As mentioned above we have now edited the Discussion over its different subtopics and added more specific examples, including related to LTP processes (starting on p.25, line 604) and other mechanisms (p.31, line 792-808), that may support and complement the findings of the current study.

Adaptation and habituation are not discussed, since as mentioned earlier the data do not support the expression of those phenomena in the current study.

Reviewer #4 (Remarks to the Author):

This is an extremely exciting manuscript that seeks to combine MRS, a methodological strength, with other neuroimaging in the context of a motor learning paradigm. The repeat-measures/longitudinal design is a real strength and the findings are intriguing. This paper describes important work that will be of interest to a broad spectrum of scientists, from neuroscientists to imaging scientists to neurochemists and motor neurologists. Where the results perhaps fall short of being fully compelling (and where the paper would be improved with additional experiments) is in the degree of controls adopted.

R4.1. Additional controls are needed to demonstrate effects. For example, in Figure 2A there is an increase in Glu around the motor learning paradigm. However, there is not control arm demonstrating that there is no increase in control regions of the brain, or in subjects not engaged in a motor learning task.

- It would also be important to show (in Supplementary) the same plots for major metabolites (e.g. NAA, Cr?) and Glu-like minor metabolites (Gln, ml ?), to address the extent to which the effects seen are Glu-specific.

We thank the Reviewer for raising these important aspects. We aimed to address all these concerns by implementing the following procedures:

1) We have added within-group control conditions by:

a. Adding additional ROI in the posterior parietal cortex (PCC) as a control region in the functional connectivity and VBM analyses. Importantly, the PCC has been used before as a control region in motor learning studies (Doyon et al. 2018; Nettekoven et al. 2022; Stagg et al. 2014). Please see the Methods section (p.9, lines 256-260) and the relevant Results sections (pp.18-22). Please also note that in order to reduce the overall number of comparisons and associations, we only addressed the PCC functional and structural metrics in the learning group, and where statistically significant results were observed to serve as a control condition for those effects.

b. Following the Reviewer's and another Reviewer's comments we examined whether there were significant changes in other major or Glu-like minor metabolites, i.e., NAA and Gln, respectively, following learning. No significant changes were observed in NAA ($p=.119$) or Gln ($p=.602$) following learning (see **Supplementary Figure 2** and the Results section p.13, line 378-380).

2) While the study was originally conduct as a within-subject repeated measures experiment in which each participant serves as her/his own control, we have followed the Reviewer's request and recruited a Control group of participants not engaging in motor learning task as the Reviewer suggested (please see the *Experimental Protocol* in p.4, line 107, and also throughout the manuscript's sections). We have managed to recruit twenty-one additional participants to serve as a Control group. It is also important to note, that all the significant effects that were observed in the original experimental group were between-subjects (i.e., correlations) and not at the group-level (including post-learning changes in Glu that were not statistically significant).

R4.2. In general, while the sample size is relatively good for a multi-scan MRS study, there are so many variables in play (pre, post, changes, metabolites etc) that it is difficult to know how strongly the stated hypotheses were fixed before the study began. For publication in a journal like NC, I would expect to see a full replication (perhaps powered for one-tailed statistics). I wrote this comment even before I realized the full range of neuroimaging performed (including

resting-state MRI, which has almost infinite potential metrics). I appreciate that replication is a high bar, but that is what I think is required here.

While resting-state fMRI has indeed many potential metrics, in this study we have focused on a single metric, namely the ROI-to-ROI connectivity between our region of interest (M1) and a predefined region, the putamen, which is one of the most important regions to participate in motor skill consolidation, as highlighted in the Introduction (p.3, line 76). Somewhat surprisingly, the number of tests and comparisons actually made is not large at all, and with the FDR corrections in place, our reported results are quite focused (i.e., this study was anything but a “fishing expedition”, with well-formulated hypotheses going in based on previous animal and human studies).

We have given serious consideration and replied extensively to all Reviewer requests, including recruiting a completely new group, and running a wide range of new “control” analyses (ROIs, other metabolites, other within-group comparisons), all of which have required several intensive months of work and all of which support our findings. Given our compliance so far, we feel a full replication of a study is truly a high bar to meet – perhaps even unfairly so, given the extent of this work, even with the high scientific standards of Nat Comms. We are also unaware of any Nat Comms publishing prerequisite of such self-replication. We hope the Reviewer comes to see eye-to-eye with us on this.

R4.3. This goes especially for results relating neurochemical change to structural change. Where there is no global change between timepoints in either axis and a correlation where some people reduce e.g. GABA and reduce tissue volume and others increase both, there is real concern about the independence of measures. the tissue-corrected GABA results and the structural measure presumably make use of the same T1 images. In replicating this result, it would be important to repeat the T1, once for tissue correction and once for volume measures. Otherwise the variables correlated are not statistically independent and noise in the segmentation will contribute (perhaps lightly) to noise in the metabolite measures. An alternative way to show this is a non-issue is to plot the same results without tissue correction or using a Cr-referenced measure.

We understand the Reviewer’s concern and therefore performed additional examination of the correlation between the changes in GABA 30 minutes following learning (which were statistically significant) and changes in the M1 GM volume. As the Reviewer suggested we examined GABA changes as either without tissue correction (using GABA signal) or using a tCr-referenced measure. Please note, that both analyses yielded significant results, similar to when using tissue-corrected values (please see Results section p.21 line 501-505, and **Supplementary Figure 3**).

R4.4. That a short learning paradigm results in a meaningful structural changes detectable by structural imaging is a challenging idea... that such a change is positive in some subjects and negative in others and still is a meaningful response to the training is even harder. What is the proposed reason for this range?

We thank the Reviewer for this question. Please note that we have not demonstrated that the paradigm resulted in meaningful structural changes (there was no significant group-level change in GM volume), only that the extent of structural changes was related to the extent of GABA changes. We also presented and highlighted that the extent of GM change was not related to overnight changes in performance.

With that being said, there is increasing evidence that structural changes may take place very shortly following motor skill learning, and previous works have showed MRI-derived structural changes following short learning paradigms, as early as minutes-hours following the learning session. We have

examples for this phenomenon in the specific subtopic in the Discussion section (please see p.30, line 745).

Doyon, J., E. Gabbitov, S. Vahdat, O. Lungu, and A. Boutin. 2018. "Current Issues Related to Motor Sequence Learning in Humans." Current Opinion in Behavioral Sciences 20:89–97. doi: <https://doi.org/10.1016/j.cobeha.2017.11.012>.

Nettekoven, Caroline, Leah Mitchell, William T. Clarke, Uzay Emir, Jon Campbell, Heidi Johansen-Berg, Ned Jenkinson, and Charlotte J. Stagg. 2022. "Cerebellar GABA Change during Visuomotor Adaptation Relates to Adaptation Performance and Cerebellar Network Connectivity: A Magnetic Resonance Spectroscopic Imaging Study." The Journal of Neuroscience : The Official Journal of the Society for Neuroscience 42(41):7721–32. doi: 10.1523/JNEUROSCI.0096-22.2022.

Stagg, Charlotte J., Velicia Bachtiar, Ugwechi Amadi, Christel A. Gudberg, Andrei S. Ilie, Cassandra Sampaio-Baptista, Jacinta O'Shea, Mark Woolrich, Stephen M. Smith, Nicola Filippini, Jamie Near, and Heidi Johansen-Berg. 2014. "Local GABA Concentration Is Related to Network-Level Resting Functional Connectivity" edited by J. C. Culham. ELife 3:e01465. doi: 10.7554/eLife.01465.

REVIEWERS' COMMENTS

Reviewer #1 (Remarks to the Author):

The authors have appropriately revised the manuscript. I commend them on a very interesting study.

Reviewer #2 (Remarks to the Author):

The authors have invested much work in addressing the questions and comments I and other reviewers had. This is an excellent work and I have no more comments to add, and recommend publication.

Reviewer #3 (Remarks to the Author):

I hereby would like to thank the authors for addressing all the questions I had and responding thoroughly to all the comments.

Their paper was already very good in my opinion and was further improved. This work paves the way to a better understanding of memory consolidation and plasticity mechanisms using MRS techniques. I really appreciated the quality of the experiments and the very insightful discussion provided by authors.

Reviewer #4 (Remarks to the Author):

Authors have been responsive to the prior review, and have undertaken substantial additional work, including recruiting and scanning an additional control cohort. They have not, as suggested replicated the main effects of the study, arguing that this is an excessive burden. I will leave that question to the editor.

The response to questions around the metabolite changes seen with learning somewhat misses the point - why would the same learning process result in a metabolite going up in some and down in others? Poor learning would be expected to associate with no change and good learning with a change.

This remains an interesting study.

Final Revision

Reviewer #4 (Remarks to the Author):

Authors have been responsive to the prior review, and have undertaken substantial additional work, including recruiting and scanning an additional control cohort. They have not, as suggested replicated the main effects of the study, arguing that this is an excessive burden. I will leave that question to the editor.

The response to questions around the metabolite changes seen with learning somewhat misses the point - why would the same learning process result in a metabolite going up in some and down in others? Poor learning would be expected to associate with no change and good learning with a change.

This remains an interesting study.

We thank the Reviewer for her/his comments. Regarding the question *why would the same learning result in opposite metabolic responses across different participants*, while we can only speculate on a potential reason for this phenomenon, please note that such a bidirectional response has also been previously reported following other types of learning (i.e., perceptual)^{1,2} as well as in response to non-invasive brain stimulation in M1^{3,4}. Those observations, taken together with the findings of the current study, may suggest that the direction of the neurochemical cortical response to external or internal stimuli across different individuals, may represent an individual trait, which may underlie or contribute to the inter-individual differences in learning capacity and potential. However, this theory needs further empirical validation.

We have added a reference on this matter in the text as follows (p.9, lines 269-277):

“Furthermore, the observation that the same stimulus (i.e., MSL task) resulted in opposite metabolic responses across different participants, follows previous reports on such an opposite response across different individuals (even if a significant group-level effect was observed) following other types of learning (i.e., perceptual)^{31,32}, as well as in response to non-invasive brain stimulation in M1^{33,34}. Those observations, taken together with the findings of the current study, may suggest that the direction of the neurochemical cortical response to external or internal stimuli across different individuals, may represent an individual trait, which may underlie or contribute to the inter-individual differences in learning capacity and potential. However, this possibility needs further direct empirical examination.”

1. Shibata K, Sasaki Y, Bang JW, Walsh EG, Machizawa MG, Tamaki M, Chang LH, Watanabe T. Overlearning hyperstabilizes a skill by rapidly making neurochemical processing inhibitory-dominant. *Nat Neurosci*. 2017 Mar;20(3):470-475. doi: 10.1038/nn.4490. Epub 2017 Jan 30.
2. Bang JW, Shibata K, Frank SM, Walsh EG, Greenlee MW, Watanabe T, Sasaki Y. Consolidation and reconsolidation share behavioral and neurochemical mechanisms. *Nat Hum Behav*. 2018 Jul;2(7):507-513. doi: 10.1038/s41562-018-0366-8. Epub 2018 Jul 9. PMID: 30505952; PMCID: PMC6258036.

3. Kim S, Stephenson MC, Morris PG, Jackson SR. tDCS-induced alterations in GABA concentration within primary motor cortex predict motor learning and motor memory: a 7 T magnetic resonance spectroscopy study. *Neuroimage*. 2014 Oct 1;99:237-43. doi: 10.1016/j.neuroimage.2014.05.070. Epub 2014 Jun 3. PMID: 24904994; PMCID: PMC4121086.

4. Stagg CJ, Bachtiar V, Johansen-Berg H. The role of GABA in human motor learning. *Curr Biol*. 2011 Mar 22;21(6):480-4. doi: 10.1016/j.cub.2011.01.069. Epub 2011 Mar 3. PMID: 21376596; PMCID: PMC3063350.